# Extended field-of-view ultrathin microendoscopes for high-resolution two-photon imaging with minimal invasiveness

Andrea Antonini[1,2†], Andrea Sattin[1,3,4†], Monica Moroni[4,5,6], Serena Bovetti[1], Claudio Moretti[1,3], Francesca Succol[1], Angelo Forli[1,4], Dania Vecchia[1,4], Vijayakumar P Rajamanickam[1,2,7], Andrea Bertoncini[7], Stefano Panzeri[4,5], Carlo Liberale[2,7], Tommaso Fellin[1,4*]

[1]Optical Approaches to Brain Function Laboratory, Istituto Italiano di Tecnologia, Genova, Italy; [2]Nanostructures Department, Istituto Italiano di Tecnologia, Genova, Italy; [3]University of Genova, Genova, Italy; [4]Neural Coding Laboratory, Istituto Italiano di Tecnologia, Genova and Rovereto, Italy; [5]Neural Computation Laboratory, Center for Neuroscience and Cognitive Systems @UniTn, Istituto Italiano di Tecnologia, Rovereto, Italy; [6]Center for Mind and Brain Sciences (CIMeC), University of Trento, Rovereto, Italy; [7]Biological and Environmental Science and Engineering Division (BESE), King Abdullah University of Science and Technology (KAUST), Thuwal, Saudi Arabia

*For correspondence:
tommaso.fellin@iit.it

†These authors contributed equally to this work

Competing interests: The authors declare that no competing interests exist.

**Abstract** Imaging neuronal activity with high and homogeneous spatial resolution across the field-of-view (FOV) and limited invasiveness in deep brain regions is fundamental for the progress of neuroscience, yet is a major technical challenge. We achieved this goal by correcting optical aberrations in gradient index lens-based ultrathin ($\leq$500 µm) microendoscopes using aspheric microlenses generated through 3D-microprinting. Corrected microendoscopes had extended FOV (*eFOV*) with homogeneous spatial resolution for two-photon fluorescence imaging and required no modification of the optical set-up. Synthetic calcium imaging data showed that, compared to uncorrected endoscopes, *eFOV*-microendoscopes led to improved signal-to-noise ratio and more precise evaluation of correlated neuronal activity. We experimentally validated these predictions in awake head-fixed mice. Moreover, using *eFOV*-microendoscopes we demonstrated cell-specific encoding of behavioral state-dependent information in distributed functional subnetworks in a primary somatosensory thalamic nucleus. *eFOV*-microendoscopes are, therefore, small-cross-section ready-to-use tools for deep two-photon functional imaging with unprecedentedly high and homogeneous spatial resolution.

## Introduction

The amount of information carried by neural ensembles and the impact that ensemble activity has on signal propagation across the nervous system and on behavior critically depend on both the information and tuning properties of each individual neuron and on the structure of correlated activity, either at the level of correlations between each pair of neurons or at the whole network level (*Ni et al., 2018*; *Salinas and Sejnowski, 2001*; *Shahidi et al., 2019*; *Shamir and Sompolinsky, 2006*; *Panzeri et al., 1999*). To study neuronal population coding, it is thus essential to be able to measure with high-precision, large signal-to-noise-ratio (SNR), and without distortions the activity of individual neurons and the relationship between them (*Aharoni et al., 2019*). Two-photon imaging makes it possible to record the activity of many hundreds of individual neurons simultaneously and provides reliable measures of correlated neuronal events (*Yang et al., 2016*; *Kazemipour et al.,*

2019; *Villette et al., 2019*; *Runyan et al., 2017*; *Rumyantsev et al., 2020*). Light scattering within the brain, however, strongly affects the propagation of excitation and emission photons, making effective imaging increasingly difficult with tissue depth (*Ji et al., 2010*; *Wang et al., 2014*; *Helmchen and Denk, 2005*). Various strategies have been developed to improve imaging depth in multi-photon fluorescence microscopy (*Kobat et al., 2009*; *Kobat et al., 2011*; *Theer et al., 2003*; *Horton et al., 2013*; *Ji et al., 2012*; *Mittmann et al., 2011*; *Tischbirek et al., 2015*; *Birkner et al., 2017*), allowing the visualization of regions 1–1.6 mm below the brain surface. However, deeper imaging requires the use of implantable microendoscopic probes or chronic windows, which allow optical investigation of neural circuits in brain regions that would otherwise remain inaccessible (*Jung and Schnitzer, 2003*; *Flusberg et al., 2008*; *Barretto et al., 2009*; *Resendez et al., 2016*; *Bocarsly et al., 2015*).

A critical barrier to progress is the lack of availability of microendoscopic devices with small cross-sections that maintain cellular resolution across a large FOV, to allow high-resolution and high SNR two-photon population imaging on a large number of neurons while minimizing tissue damage. Preserving short- and long-range connectivity is fundamental to study physiological network dynamics in brain circuits. For example, severing thalamocortical fibers alters low-frequency spontaneous oscillations in the thalamocortical loop (*Lemieux et al., 2014*; *Lemieux et al., 2015*). Current microendoscopes for deep imaging are frequently based on gradient index (GRIN) rod lenses, which typically have diameter between 0.35–1.5 mm and are characterized by intrinsic optical aberrations (*Barretto et al., 2009*). These aberrations are detrimental in two-photon imaging because they decrease the spatial resolution and lower the excitation efficiency, especially in the peripheral parts of the FOV. This leads to uneven SNR and spatial resolution across the FOV and therefore restricted effective FOV (*Wang and Ji, 2013*; *Ji et al., 2016*). This is a serious issue when measuring correlated neuronal activity, because signals sampled at the periphery of the FOV will be more contaminated by neuropil or by activity of neighboring cells compared to signals sampled near the optical axis. Optical aberrations in GRIN microendoscopes can be corrected with adaptive optics which, however, requires significant modification of the optical path (*Wang and Ji, 2013*; *Wang and Ji, 2012*; *Bortoletto et al., 2011*) and may limit the temporal resolution of functional imaging over large FOVs (*Wang and Ji, 2013*). Alternatively, the combination of GRIN lenses of specific design with plano-convex lenses within the same microendoscopic probe has been used to increase the Numerical Aperture (NA) and to correct for aberrations on the optical axis (*Barretto et al., 2009*). However, technical limitations in manufacturing high-precision free-form optics with small lateral dimensions have so far prevented improvements in the performances of GRIN microendoscopes with lateral diameter <1 mm using corrective optical microelements (*Matz et al., 2015*).

Here, we report the design, development, and characterization of a new approach to correct aberrations and extend the FOV in ultrathin GRIN-based endoscopes using aspheric lenses microfabricated with 3D micro-printing based on two-photon lithography (TPL) (*Liberale et al., 2010*). These new endoscopic probes are ready-to-use and they require no modification of the optical set-up. We applied *eFOV*-microendoscopes to study the encoding of behavioral state-dependent information in a primary somatosensory thalamic nucleus of awake mice in combination with genetically-encoded calcium indicators.

## Results

### Optical simulation of *eFOV*-microendoscopes

Four types (type I-IV) of *eFOV*-microendoscopes of various length and cross section were designed, all composed of a GRIN rod, a glass coverslip and a microfabricated corrective aspheric lens (*Figure 1*). One end of the GRIN rod was directly in contact with one side of the glass coverslip while the corrective aspheric lens was placed on the other side of the coverslip. GRIN rods and corrective lenses were different in each of the four types of *eFOV*-microendoscopes (lateral diameter, 0.35–0.5 mm; length, 1.1–4.1 mm; all 0.5 NA). The glass coverslip was 100 μm thick for type I, III, IV *eFOV*-microendoscopes and 200 μm thick for type II *eFOV*-microendoscopes. This design did not require additional cannulas or clamping devices (*Kim et al., 2012*; *Jung et al., 2004*) that would increase the lateral size of the microendoscope assembly or reduce its usable length, respectively.

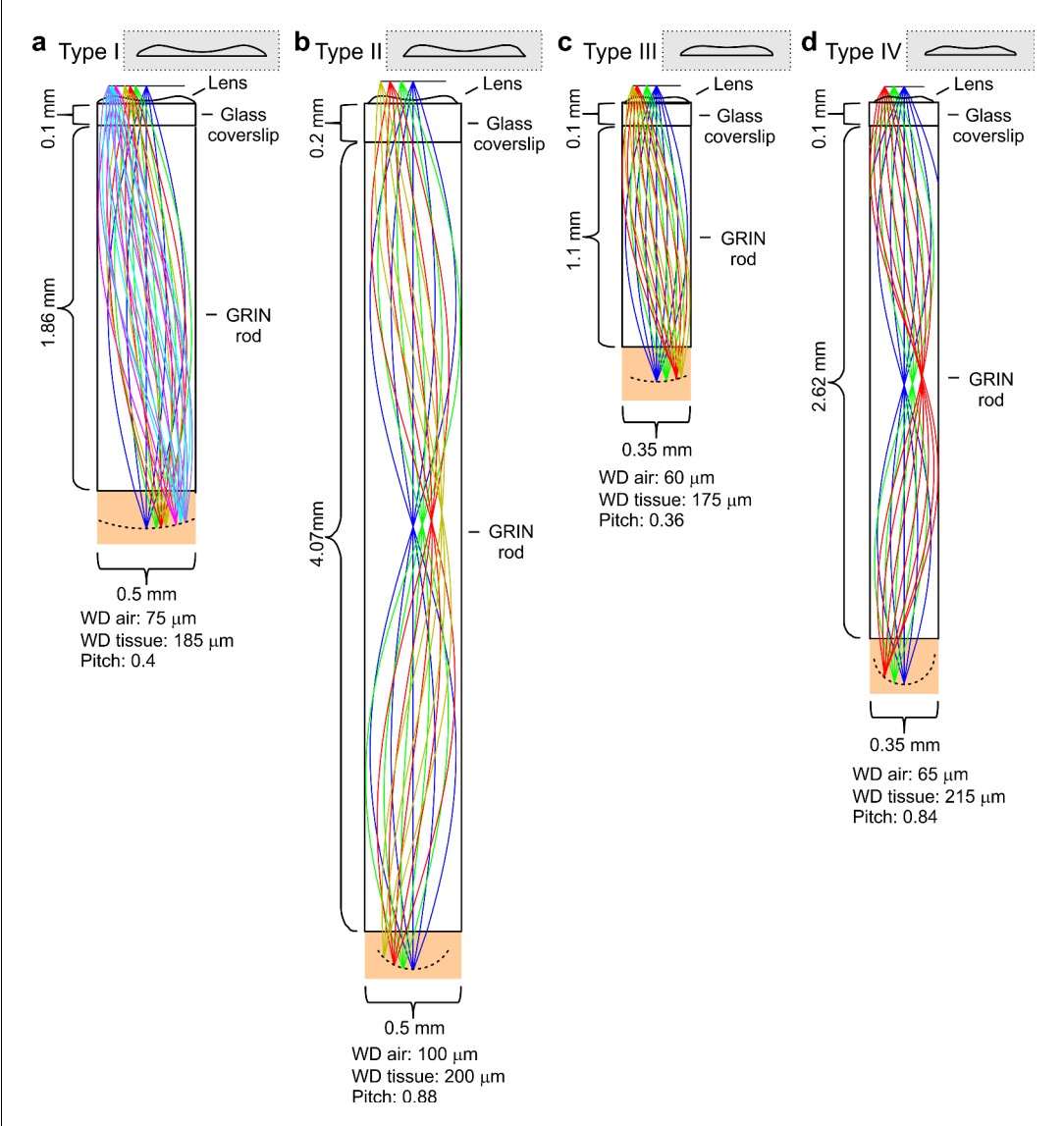

**Figure 1.** Optical design of *eFOV*-microendoscopes. (**a-d**) Ray-trace simulations for the four different *eFOV*-microendoscopes (type I-IV). The insets show the profiles of corrective polymeric lenses used in the different *eFOV*-microendoscopes. For each *eFOV*-microendoscope, it is specified the thickness of the coverslip, the length, the diameter of the GRIN rod, the pitch of the GRIN rod, and the working distance, in air or in tissue, at which the simulated correction of aberrations was performed. See also *Supplementary file 1 - Table 1*.

For each type of GRIN rod used in the *eFOV*-microendoscopes, ray-trace simulations were performed at λ = 920 nm and determined the profile of the aspheric lens (*Figure 1*) that corrected optical aberrations and maximized the FOV (*Figure 2*). In the representative case of type I *eFOV*-microendoscopes, the simulated corrective lens had a diameter of 0.5 mm, height <40 µm, and the coefficients used in *Equation (1)* (see Materials and methods) to define the lens profile are reported in *Supplementary file 1 - Table 1*. For this type of *eFOV*-microendoscope, the simulated point-spread-function (PSF) at incremental radial distances from the optical axis (up to 200 µm) showed that the Strehl ratio of the system was >80% (i.e. a diffraction-limited condition is achieved according to the Maréchal criterion [*Smith, 2008*]) at a distance up to ~165 µm from the optical axis with the corrective lens, while only up to ~70 µm for the same optical system without the corrective lens (*Figure 2a*). This improvement led to a ~5 times increase in the area of the diffraction-limited FOV. *Figure 2b–d* reports the Strehl ratio for simulated uncorrected and corrected type II-IV *eFOV*-microendoscopes. Simulations showed that the area of the FOV was ~2–9 times larger for these other

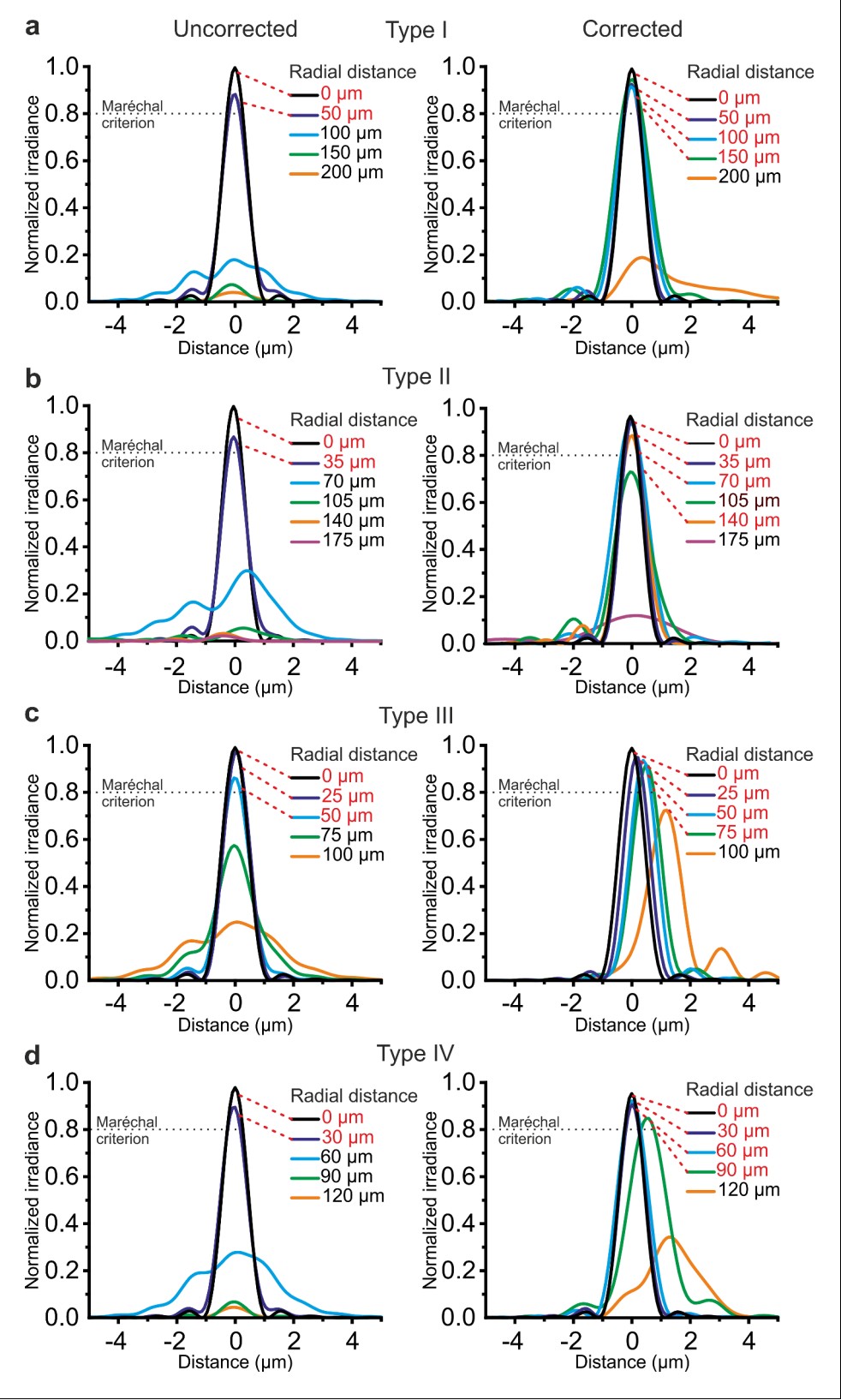

**Figure 2.** Corrective lenses improve the simulated optical performance of ultrathin microendoscopes. (**a**) Simulated diffraction PSFs to assess the Strehl ratio of the designed microendoscope (type I microendoscopes) without the corrective lens (uncorrected, left) and with the corrective lens (corrected, right). PSFs are shown color-

*Figure 2 continued on next page*

*Figure 2 continued*

coded according to their radial distance from the optical axis. The black dotted line represents the diffraction-limited condition, which was set at 80% (Maréchal criterion). Distances written in red indicate the radial positions at which the maximal normalized irradiance of the corresponding PSF was >80%. (**b–d**) Same as in (**a**) for type II (**b**), type III (**c**), and type IV (**d**) microendoscopes. See also *Figure 2—figure supplements 1–4*.

The online version of this article includes the following source data and figure supplement(s) for figure 2:

**Figure supplement 1.** Aberration correction improves the simulated PSF in the peripheral portions of the FOV in *eFOV*-microendoscopes.

**Figure supplement 2.** Simulated spatial resolution is improved in *eFOV*-microendoscopes.

**Figure supplement 2—source data 1.** Simulated values of the axial and lateral spatial resolution as a function of radial distance from the center of the FOV in uncorrected and corrected microendoscopes.

**Figure supplement 3.** Corrective lenses enlarge the simulated volume that can be scanned during 3D imaging.

**Figure supplement 3—source data 1.** Simulated Strehl ratio as a function of the lateral and axial displacement for type I-IV uncorrected and corrected microendoscopes.

**Figure supplement 4.** Chromatic aberrations in *eFOV*-microendoscopes.

**Figure supplement 4—source data 1.** Simulated Strehl ratio as a function of wavelength.

---

types of *eFOV*-microendoscopes, compared to microendoscopes without the corrective lens. We evaluated the simulated excitation PSF for the four types of microendoscopes (*Figure 2—figure supplements 1–2*). We found that the axial dimension of the PSF in lateral portions of the FOV remained smaller and more similar to the axial dimension of the PSF in the center of the FOV in corrected compared to uncorrected endoscopes.

The simulated focal length in the absence and presence of the corrective microlens for the four different types of microendoscopes is reported in *Supplementary file 1 - Table 2*. All simulations reported above were performed maximizing aberration correction only in the focal plane of the microendoscopes. We explored the effect of aberration correction outside the focal plane. In corrected endoscopes, we found that the Strehl ratio was >0.8 (Maréchal criterion) in a 1.7–3.7 times larger volume compared to uncorrected endoscopes (*Figure 2—figure supplement 3*). We then investigated the effect of changing wavelength on the Strehl ratio. We found that the Strehl ratio remained >0.8 within at least ±15 nm from λ = 920 nm (*Figure 2—figure supplement 4*), which covered the limited bandwidth of our femtosecond laser.

## Fabrication of *eFOV*-microendoscopes

Corrective lenses were experimentally fabricated using TPL (*Liberale et al., 2010*; *Figure 3—figure supplement 1a,b*) and plastic molding replication (*Schaap and Bellouard, 2013*) directly onto the glass coverslip (see Materials and methods). Experiments and optical characterization were performed using lens replica only. Fabricated lenses had profile largely overlapping with the simulated one (*Figure 3—figure supplement 1c*). The corrective lens was aligned to the appropriate GRIN rod using a customized optomechanical set-up (*Figure 3—figure supplement 2a,b*) to generate *eFOV*-microendoscopes. To experimentally validate the optical performance of fabricated *eFOV*-microendoscopes, we first coupled them with a standard two-photon laser-scanning system using a customized mount (*Figure 3a,b* and *Figure 3—figure supplement 2c,d*). We initially measured the on-axis spatial resolution by imaging subresolved fluorescence beads (diameter: 100 nm) at 920 nm. We found that *eFOV*-microendoscopes had similar on-axis axial resolution compared to uncorrected probes (*Figure 3c,e,g,i* left and *Supplementary file 1 - Table 3*). Given that most aberrations contribute to decrease optical performance off-axis, we repeated the measurements described above and measured the axial and lateral resolution at different radial distances from the center of the FOV. As expected by ray-trace simulations (*Figure 2*), *eFOV*-microendoscopes displayed higher and more homogeneous axial resolution in a larger portion of the FOV (*Figure 3c,e,g,i* left and *Figure 3—figure supplement 3*). Defining the effective FOV as that in which $FWHM_z$ <10 μm (dotted line in *Figure 3c,e,g,i* left), we found ~3.2–9.4 folds larger effective FOV in corrected microendoscopes compared to uncorrected probes (*Supplementary file 1 - Table 3*).

To visualize the profile of fluorescence intensity across the whole diameter of the FOV for both uncorrected and corrected probes, we used a subresolved thin fluorescent layer (thickness: 300 nm) as detailed in *Antonini et al., 2014*. *Figure 3d,f,h,j* shows the x, z projections of the z-stack of the

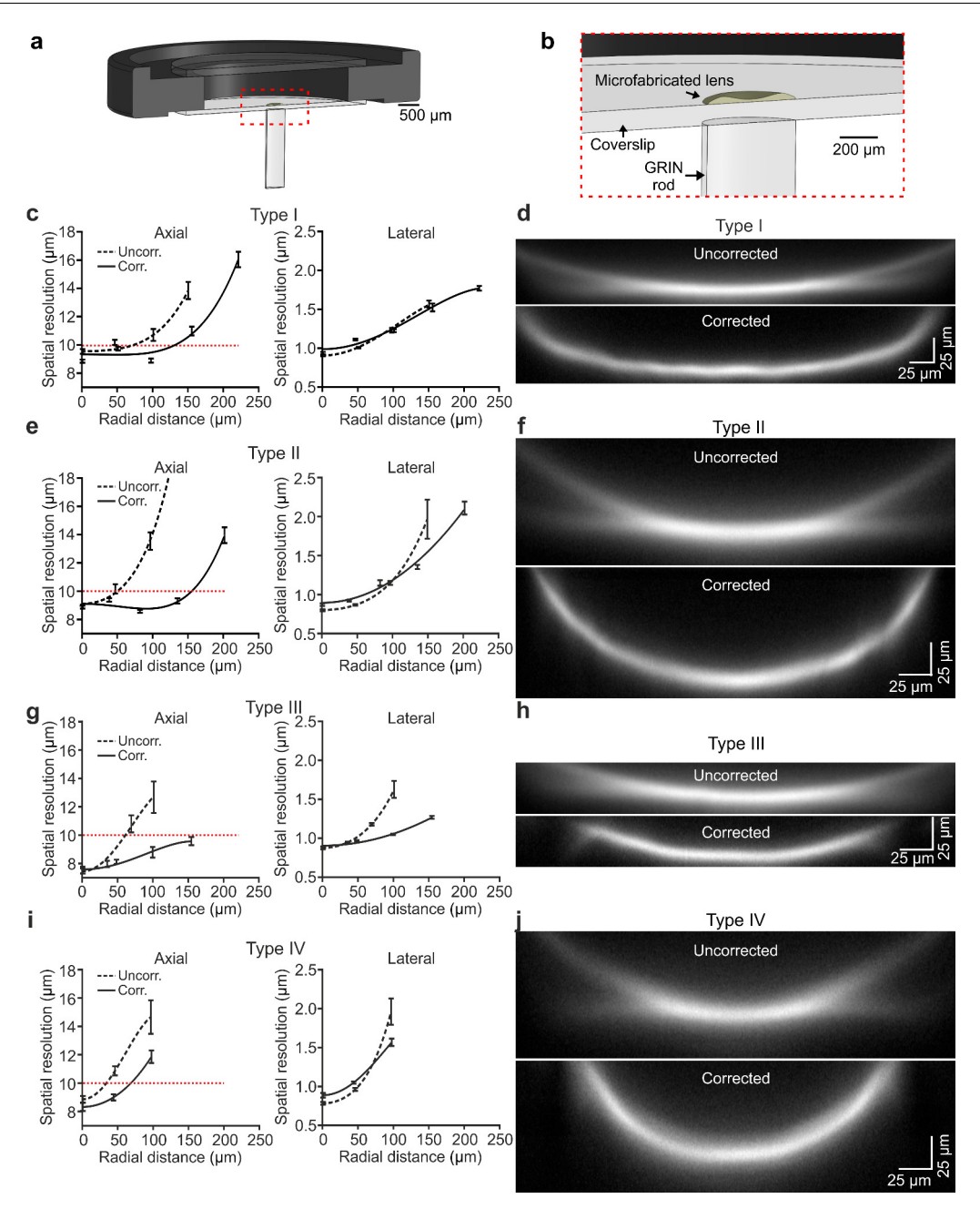

**Figure 3.** Optical characterization shows enlarged effective FOV in corrected ultrathin microendoscopes. (**a**) Schematic of the *eFOV*-microendoscope mount for head implant. The GRIN rod is glued to one side of the glass coverslip, the microfabricated polymer lens to the other side of the coverslip. The coverslip is glued on a circular metal ring that facilitates fixation on the animal's skull. (**b**) Zoom in of the portion highlighted with the red dotted line in (**a**). (**c**) Axial (left) and lateral (right) spatial resolution (see Materials and methods for definition) evaluated using subresolved fluorescent beads (diameter: 100 nm) as a function of the radial distance from the center of the FOV for type I uncorrected (dashed line) and corrected (solid line) microendoscopes. Points represent values obtained by averaging at least eight measurements from three different probes (see *Supplementary file 1 - Table 1*), while error bar represents standard deviation (sd). Lines are quartic functions fitting the data. The red dashed line indicates a threshold value (10 µm) to define the limit of the effective FOV. (**d**) x,z projections (x, horizontal direction; z, vertical direction) of a z-stack of two-photon laser-scanning images of a subresolved fluorescent layer (thickness: 300 nm) obtained using a type I *eFOV*-microendoscope, without (uncorrected, top) and with (corrected, bottom) the microfabricated corrective lens. $\lambda_{exc}$ = 920 nm. (**e,f**) Same as in (**c,d**) for type II *eFOV*-microendoscopes. (**g,h**) Same as in (**c,d**) for type III *eFOV*-microendoscopes. (**i,j**) Same as in (**c,d**) for type IV *eFOV*-microendoscopes. See also *Figure 3—figure supplements 1–7* and *Supplementary file 1 - Table 3*.

The online version of this article includes the following source data and figure supplement(s) for figure 3:

*Figure 3 continued on next page*

*Figure 3 continued*

**Source data 1.** Experimental values of the axial and lateral spatial resolution as a function of radial distance from the center of the FOV in uncorrected and corrected microendoscopes.

**Figure supplement 1.** Polymer lens fabrication using 3D micro-printing based on two-photon lithography.

**Figure supplement 2.** Set-ups for the assembly and characterization of *eFOV*-microendoscopes.

**Figure supplement 3.** Aberration correction improves the PSF in the peripheral portions of the FOV in *eFOV*-microendoscopes.

**Figure supplement 4.** Field curvature in *eFOV*-microendoscopes.

**Figure supplement 4—source data 1.** Magnification correction factor as a function of the radial position in uncorrected and corrected type II microendoscopes.

**Figure supplement 5.** Corrected microendoscopes have extended effective FOV.

**Figure supplement 6.** Hippocampal imaging with type I and type III *eFOV*-microendoscopes.

**Figure supplement 7.** VPM imaging with type II *eFOV*-microendoscopes.

subresolved fluorescent layer for uncorrected and corrected type I-IV microendoscopes. In agreement with the measurements of spatial resolution using subresolved fluorescent beads (*Figure 3c,e, g,i*), *eFOV*-microendoscopes displayed higher intensity and smaller FWHM$_z$ in peripheral portions of the FOV compared to uncorrected probes (*Figure 3d,f,h,j*). *eFOV*-microendoscopes were characterized by a curved FOV and this distortion was evaluated using a fluorescent ruler (*Figure 3—figure supplement 4*) and corrected for in all measurements of spatial resolution (*Figure 3* and *Supplementary file 1 - Table 3*). The ability of *eFOV*-microendoscopes to image effectively larger FOV compared to uncorrected probes was also confirmed in biological tissue by imaging neurons expressing the green fluorescence protein (GFP) in fixed brain slices (*Figure 3—figure supplement 5*).

## Validation of *eFOV*-microendoscopes for functional imaging in subcortical regions

To validate *eFOV*-microendoscopes performance for functional measurements in vivo, we first expressed the genetically-encoded calcium indicator GCaMP6s in the mouse hippocampal region in anesthetized mice (*Figure 3—figure supplement 6*) and in the ventral posteromedial nucleus of the thalamus (VPM), a primary sensory thalamic nucleus, in awake head-restrained mice (*Figure 3—figure supplement 7*). To this aim, we injected either adenoassociated viruses (AAVs) carrying a flex GCaMP6s construct together with AAVs carrying a Cre-recombinase construct under the CamKII promoter in the hippocampus of wild type mice (*Figure 3—figure supplement 6a-a$_2$*) or AAV carrying a floxed GCaMP6s in the VPM (*Figure 3—figure supplement 7a,b*) of Scnn1a-Cre mice (*Madisen et al., 2010*), a mouse line which expresses Cre in a large subpopulation of VPM neurons. These experimental protocols resulted in the labeling of CA1 hippocampal and VPM neurons, respectively (*Figure 3—figure supplements 6–7*). We imaged spontaneous activities in the CA1 hippocampal region with type I *eFOV*-microendoscopes (*Figure 3—figure supplement 6b-c$_1$*) and type III *eFOV*-microendoscopes (*Figure 3—figure supplement 6d-e$_1$*) or spontaneous activities in the VPM with the longer type II *eFOV*-microendoscopes (*Figure 3—figure supplement 7b–d*). Tens to hundreds of active ROIs *per* single FOV were identified and could be imaged using *eFOV*-microendoscopes (*Figure 3—figure supplements 6–7*).

## Higher SNR and more precise evaluation of pairwise correlation in *eFOV*-microendoscopes

To establish a quantitative relationship between the improved optical properties of *eFOV*-microendoscopes and their potential advantages for precisely detecting neuronal activity, we generated two-photon imaging *t*-series using synthetic GCaMP data. This approach allowed us to compare results of the simulation of calcium data for both uncorrected and corrected endoscopic probes with the known ground truth of neuronal activity. Simulated neuronal activity within a volumetric distribution of cells was generated according to known anatomical and functional parameters of the imaged region (the VPM in this case) and established biophysical properties of the indicator (*Chen et al., 2013*; *Dana et al., 2019*) (see Materials and methods). *t*-series were generated by sampling simulated neuronal activity across an imaging focal surface resembling the experimental data obtained using the representative case of type II GRIN lenses for both *eFOV*-microendoscopes and

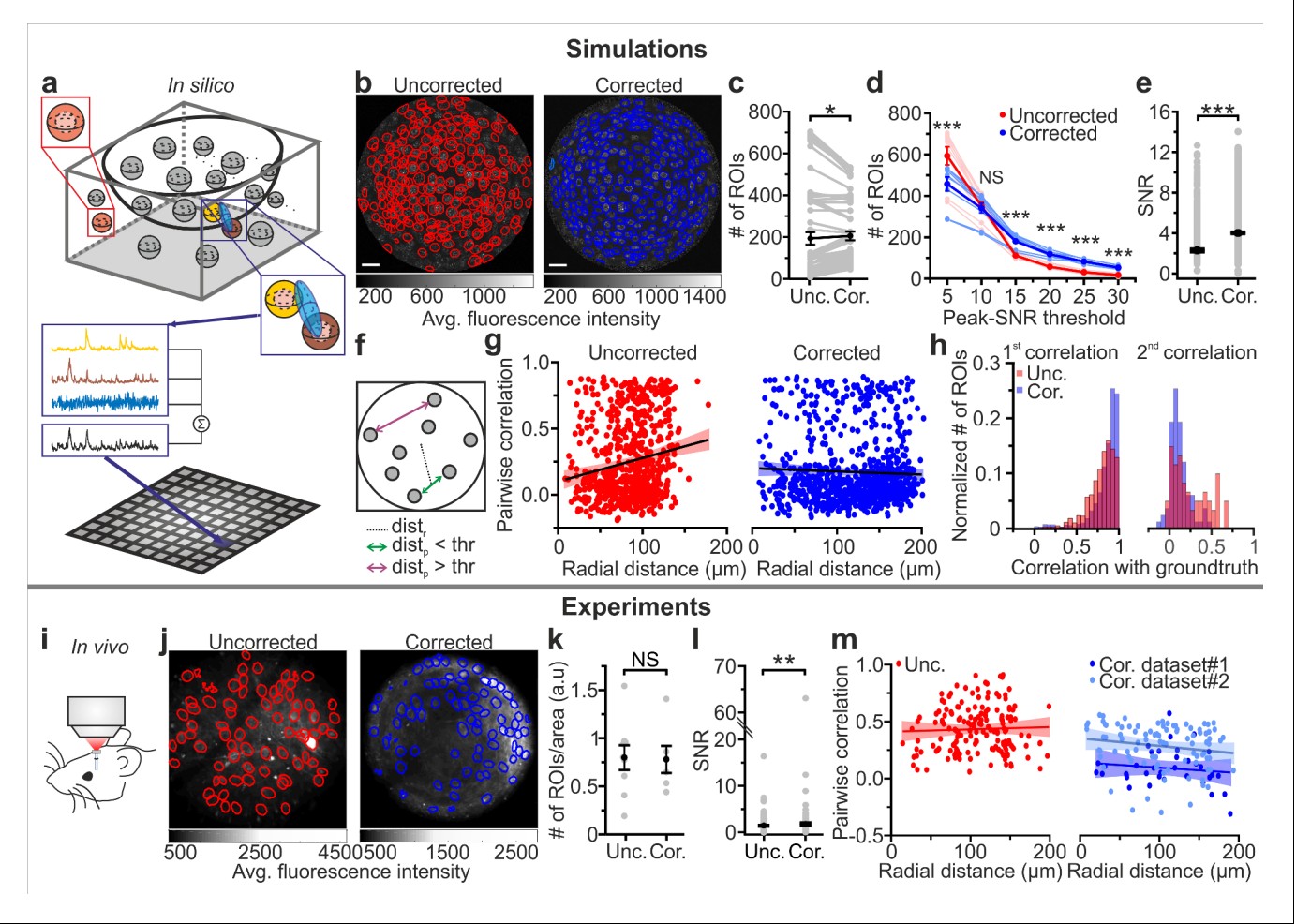

**Figure 4.** *eFOV*-microendoscopes allow higher SNR and more accurate evaluation of pairwise correlation. (**a**) Schematic of the procedure for in silico simulation of imaging data. Neuronal activity was simulated within spheres located in a 3D volume, integrated over an elliptical PSF (blue) that was scanned on a curved FOV, projected on a 2D plane to generate artificial frames, and modulated through an intensity mask. Only voxels falling within the PSF contributed to the pixel signal (black trace). (**b**) Segmentation of in silico data for uncorrected (left, red lines indicate identified ROIs) and corrected (right, blue lines indicate identified ROIs) endoscopes. (**c**) Number of ROIs segmented in simulated FOVs for uncorrected (Unc.) and corrected (Cor.) microendoscopes. n = 54 segmentations from nine simulated FOVs, Wilcoxon signed-rank test, p=0.037. In this as well as in other figures, values from individual experiments are shown in gray, the average of all experiments in black and error bars indicate sem, unless otherwise stated. In this as well as in other figures: *p<0.05; **p<0.01; ***p<0.001; NS, not significant. (**d**) Number of segmented ROIs as a function of the peak-SNR threshold in artificial data from n = 9 simulated experiments. A two-way ANOVA with interactions showed a significant effect of peak-SNR threshold (p=1E-50) and of the interaction between peak-SNR threshold and probe type (p=1E-5) on the number of segmented ROIs, while the effect of probe type was not significant (p=0.096). (**e**) Average SNR of calcium signals under the different experimental conditions (peak-SNR threshold = 20 for the segmentation). n = 987 and 1603 ROIs for nine simulated experiments with uncorrected and corrected microendoscopes, respectively. Mann-Whitney test, p=5E-52. (**f**) Schematic representation of how radial distance (dist$_r$) and pairwise distance (dist$_p$) were calculated. (**g**), Pairwise correlation as a function of radial distance for simulated experiments with uncorrected (left) and corrected (right) microendoscopes. In this as well as other figures, lines show the linear regression of data. Shaded areas represent 95% confidence intervals. n = 738 and n = 869 pairwise correlations for uncorrected and corrected microendoscopes, respectively, from the n = 9 simulated experiments shown in (**e**). Linear regression fit of data: slope = 0.002, permutation test, p=0 and slope = −2E-4, permutation test, p=0.27 for uncorrected and corrected microendoscopes, respectively. (**h**) Distribution of the correlation of calcium signals with the first (left) or second (right) component of the ground truth for experiments with uncorrected and corrected microendoscopes. First component: n = 987 and 1603 ROIs for uncorrected and corrected microendoscopes, respectively, from n = 9 simulated experiments. Second component: n = 62 and 85 ROIs for uncorrected and corrected microendoscopes, respectively, from n = 9 experiments. (**i**) Schematic of the experimental configuration in awake animals. (**j**) Two-photon images of VPM neurons expressing GCaMP7f showing manually identified ROIs for uncorrected (left, red lines) and corrected (right, blue lines) type II microendoscopes. (**k**) Spatial density of ROIs identified in in vivo experiments. n = 9 FOVs and 6 FOVs from three animals with uncorrected and corrected microendoscopes, respectively. Student's *t*-test, p=0.92. (**l**) SNR of segmented ROIs in in vivo recordings. n = 557 ROIs from 9 FOVs for uncorrected microendoscopes; n = 306 from 6 FOVs for corrected microendoscopes. Mann-Whitney test, p=0.0011. (**m**) Pairwise correlation as a function of radial distance for in vivo experiments. Number of pairwise

*Figure 4 continued on next page*

*Figure 4 continued*

correlations: n = 168 from 9 FOVs, n = 36 from 6 FOVs, and n = 92 from 24 FOVs for uncorrected, corrected (dataset 1), and corrected (dataset 2), respectively. Dataset two was obtained from experimental sessions performed during behavioral state monitoring as in *Figure 6*. Linear regression fit of data: slope = 0.0002, permutation test p=0.006 for uncorrected; slope = −0.0005, permutation test p=0.05 for dataset 1; slope = −0.0007, permutation test p=0.05 for dataset 2. See also *Figure 4—figure supplements 1–2*.

The online version of this article includes the following source data and figure supplement(s) for figure 4:

**Source data 1.** Results of manual segmentation: # of ROIs, SNR, and pairwise correlations for simulated and experimental data.

**Figure supplement 1.** Manual vs. automated segmentation in simulated and experimental *t*-series.

**Figure supplement 1—source data 1.** Comparison of manual vs automated segmentation methods in simulated and experimental data.

**Figure supplement 2.** Improved description of neuronal signals in *eFOV*-microendoscopes is observed in *t*-series automatically segmented with CalmAn.

**Figure supplement 2—source data 1.** Results of automated segmentation: # of ROIs, SNR, and pairwise correlations for simulated and experimental data.

---

uncorrected probes (*Figure 4a,b*). To scan the imaging focal surface, we used an excitation volume which resembled the aberrated and aberration-corrected PSFs experimentally measured for uncorrected and *eFOV*-microendoscopes, respectively (*Figure 3*, see Materials and methods). Fluorescence traces were extracted from artificial *t*-series and compared between the uncorrected and corrected case. On average, a slightly larger number of ROIs could be identified in corrected probes (*Figure 4c*). Crucially, we observed a nonlinear interaction between the type of probe, the number of detected ROIs, and the SNR of the calcium traces (*Figure 4d*). Using corrected probes did not always allow the identification of a higher number of ROIs (p=0.096, two-way ANOVA with respect to probe type). Rather, the use of corrected probes allowed to segment more ROIs with high SNR, shifting the distribution of SNR across ROIs to higher mean SNR values (*Figure 4d*, p=1E-5 for ANOVA with respect to the interaction, and *Figure 4e*). Corrected endoscopes allowed to segment a smaller number of ROIs with low SNR likely because: (i) in the segmentation method we implemented, which was based on the ground truth distribution of the neurons in the simulated sample, at least two pixels belonging to a ground truth neuron were defined as a ROI; (ii) in uncorrected microendoscopes, the axial PSF largely increased as a function of the radial distance. The enlarged axial PSF in the lateral portions of the FOV augmented the probability of sampling voxels belonging to multiple neurons located at different z positions. Once projected in the 2D plane, the contribution of multiple neurons located at different z positions increased the probability of having pixels belonging to ROIs. An increased axial PSF thus led to an increased number of detected ROIs; (iii) corrected endoscopes had smaller axial PSF compared to uncorrected ones in lateral portions of the FOV potentially leading to smaller number of detected ROIs.

Pairwise correlation between nearby neurons (distance between the center of neurons <20 μm) should not vary with the radial distance because in our simulations this value was constant across neurons. However, we found an artefactual increase of correlation strength with the radial distance of neuronal pairs in uncorrected endoscopes due to the cross-contamination of activity at different points generated by the larger and aberrated PSF without the corrective lens. In contrast, correlation strength remained constant in *eFOV*-microendoscopes (*Figure 4f,g*), suggesting that the PSF of the corrected probes was small enough across the FOV to decrease contamination of activity across neurons. This result is thus in agreement with the decreased spatial resolution observed in more distal parts of the FOV in uncorrected probes and the improved and homogeneous resolution across the FOV that is instead found in corrected microendoscopes (*Figure 3*). Overall, these findings suggest that signal corresponding to individual neurons could be more accurately extracted from ROIs across the FOV in corrected microendoscopes. We quantified this in synthetic data by evaluating, for each ROI, the correlation of the extracted calcium trace with the ground truth fluorescence signal generated by the simulated neuronal activity contained in that ROI (*Figure 4h* left). For those ROIs whose fluorescence dynamics were determined by more than one neuron, the correlation with the second most relevant cell was also calculated (*Figure 4h* right). We found that ROIs segmented from *eFOV*-microendoscopes displayed larger correlation with the ground truth signal of an individual neuron (*Figure 4h* left) and lower correlation with the ground truth signal of other nearby neurons (*Figure 4h* right) compared to uncorrected probes, suggesting that aberration correction allowed to

collect more precisely from single cellular emitters and with decreased cross-contamination between neurons.

We experimentally validated the results of the simulations performing functional imaging in the VPM using type II *eFOV*-microendoscopes before and after the removal of the corrective microlens in awake head-restrained mice (*Figure 4i,j*). The number of ROIs detected under the two conditions was not significantly different (*Figure 4k*). However, we found increased average SNR of calcium signals in corrected compared to uncorrected probes (*Figure 4l*), confirming also in experimental data (as it happens in simulations, *Figure 4e*) that the use of an *eFOV*-microendoscope shifts the distribution of SNR across ROIs toward having a higher proportion of ROIs with large SNR value. Moreover, the linear fit of pairwise correlations as a function of radial distance for uncorrected endoscopes had a significantly positive slope (*Figure 4m*, left panel), indicating higher pairwise correlations in lateral compared to more central portions of the FOV. For corrected endoscopes, the slope of the linear fit was not significantly different from zero (*Figure 4m*, right panel), in agreement with the analysis of the artificial calcium *t*-series (*Figure 4g*). Overall, results of simulations and experiments demonstrate that correcting optical aberrations in *eFOV*-microendoscopes enabled higher SNR and more precise evaluation of pairwise correlations compared to uncorrected probes.

To evaluate if the segmentation method could affect these results, we compared the quality of the manual segmentation method used in previous experiments with that of a standard automated algorithm (e.g. CaImAn, [*Giovannucci et al., 2019*]) by computing precision, recall, and F1 score in simulated data (*Figure 4—figure supplement 1*). We found that the automated method had recall values, on average across SNR values,<0.4 (*Figure 4—figure supplement 1a*), leading to the detection of only a minority of the total number of neurons. In contrast, the manual method tended to have higher recall across SNR threshold values. Moreover, for low SNR threshold values the automated segmentation had precision values < 0.8 in both uncorrected and corrected endoscopes (*Figure 4—figure supplement 1b*), leading to identification of ROIs which did not correspond to cells in the ground truth. In contrast, the manual segmentation method tended to have larger values of precision across SNR threshold levels. Overall, F1 scores tended to be higher for the manual segmentation method compared to the automated one for both uncorrected and corrected endoscopes (*Figure 4—figure supplement 1c*). We extended the comparison between the manual and automated segmentation methods to the real data shown in *Figure 4i–m*. We observed that in uncorrected endoscopes the automated method identified smaller number of ROIs compared to manual segmentation (*Figure 4—figure supplement 1d*). In contrast, the number of ROIs identified with the automated approach and the manual method in *t*-series acquired with the corrected endoscope were not significantly different (*Figure 4—figure supplement 1e*). One potential explanation of this finding is that the automated segmentation method more efficiently segments ROIs with high SNR compared to the manual one. Since aberration correction significantly increases SNR of fluorescent signals, the automated segmentation performed as the manual segmentation method in corrected endoscopes.

We evaluated the effect of aberration correction on the output of the analysis in the simulated and experimental data shown in *Figure 4* using an automated segmentation method (e.g. CaImAn). In simulated data, we found that using CaImAn the number of ROIs segmented in corrected endoscopes was consistently higher than in the uncorrected case across SNR thresholds (*Figure 4—figure supplement 2a*), similarly to what observed with manual segmentation (*Figure 4d*). Using CaImAn, SNR values of fluorescence events were significantly higher in corrected compared to uncorrected endoscopes (*Figure 4—figure supplement 2b*), similarly to what observed with manual segmentation (*Figure 4e*). Moreover, the slope of the linear fit of pairwise correlations as a function of radial position for uncorrected endoscopes had a significantly positive slope (*Figure 4—figure supplement 2c* left panel), indicating higher pairwise correlations in lateral compared to more central portions of the FOV. For corrected endoscopes, the slope of the linear fit was not significantly different from zero (*Figure 4—figure supplement 2c* right panel). This result is also in line with what previously observed with manual segmentation (*Figure 4g*). In experimental data, we found that using CaImAn SNR values of fluorescence events tended to be higher in corrected compared to uncorrected endoscopes (*Figure 4—figure supplement 2d*), a trend that was in line with what observed with manual segmentation (*Figure 4l*). The slope of the linear fit of pairwise correlations as a function of radial position for uncorrected endoscopes was significantly positive (*Figure 4—figure supplement 2e*), indicating higher pairwise correlations in lateral compared to more central portions of the

FOV. For corrected endoscopes, the slope of the linear fit was not significantly different from zero (*Figure 4—figure supplement 2e*). Both results are in line with what previously observed with manual segmentation (*Figure 4m*). Overall, the results of the comparison between the manual and automated segmentation methods showed that improvements introduced by aberration correction in endoscopes were observed with both the automated and manual segmentation methods.

## Spatial mapping of behavior state-dependent information in sensory thalamic nuclei in awake mice

We then focused our attention on the VPM, a key region which relays somatosensory (whisker) information to the barrel field of the primary somatosensory cortex (S1bf) through excitatory thalamocortical fibers (*Feldmeyer et al., 2013*). VPM also receives strong cortical feedback from corticothalamic axons of deep cortical layers. Cortical inputs to VPM has been proposed to strongly modulate thalamic activity. Thus, to study VPM physiology it is fundamental to preserve corticothalamic and thalamocortical connectivity. Electrophysiological recordings showed that VPM networks are modulated by whisking and behavioral state (*Urbain et al., 2015*; *Moore et al., 2015*; *Poulet et al., 2012*). However, how information about whisking and other behavioral state-dependent processes (e.g arousal, locomotion) are spatially mapped in VPM circuits at the cellular level is largely unknown. We used *eFOV*-microendoscopes to address this question.

As an important control experiment, we first confirmed that the ultrathin GRIN lenses that we used in our study (diameter ≤500 μm) did not significantly damage anatomical thalamocortical and corticothalamic connectivity, a difficult task to achieve with larger cross-section GRIN lenses or with chronic optical windows (*Figure 3—figure supplement 7e*). To this aim, we performed local co-injections in the VPM of Scnn1a-Cre mice of red retrobeads to stain corticothalamic projecting neurons with axons targeting the VPM and of an adenoassociated virus carrying a floxed GFP construct to stain thalamocortical fibers (*Figure 5a*). We evaluated the amount of thalamocortical and corticothalamic connectivity looking at the percentage of pixels displaying green and red signal in the S1bf region in endoscope-implanted vs. non-implanted mice (*Figure 5a–c*). In accordance with the known anatomy of the thalamocortical system (*Feldmeyer, 2012*), we found that the green signal was mostly localized in layer IV barrels and in layer V/VI while the red signal was largely restricted to layer VI (*Figure 5d,e*). Importantly, we found no difference in the percentage of pixels displaying green and red signals in implanted vs. non-implanted mice (*Figure 5f*).

We then used *eFOV*-microendoscopes to address the question of how information about motor behavior (e.g. locomotion and whisking) and internal states (e.g. arousal state) are mapped on VPM circuits at the cellular level. To this aim, we used *eFOV*-microendoscopes to perform GCaMP6s imaging in VPM circuits in awake head-restrained mice while monitoring locomotion, whisker mean angle, and pupil diameter (*Figure 6a–d*, see Materials and methods). We identified quiet (Q) periods, time intervals that were characterized by the absence of locomotion and whisker movements, and active (A) periods, intervals with locomotor activity, dilated pupils, and whisker movements. Active periods were further subdivided into whisking (W), whisking and locomotion (WL), and locomotion with no whisking (L). *Figure 6e* shows a histogram representing the amount of time spent in the different behavioral states. Mice whisk when they move, therefore L periods were rare. We found that Q periods showed calcium events that were sparsely distributed both across time and neurons (*Figure 6c,d*). In contrast, active periods displayed an increase in both frequency and amplitude of calcium signals across VPM neurons compared to Q periods (average frequency: $f_Q = 1.95 \pm 0.02$ Hz, $f_A = 2.22 \pm 0.02$ Hz, Student's $t$-test p=2E-74, n = 24 $t$-series from four mice; average amplitude: $A_Q = 0.137 \pm 0.005$ $\Delta F/F_0$, $A_A = 0.245 \pm 0.008$ $\Delta F/F_0$, Student's $t$-test p=2E-128, n = 24 $t$-series from four mice). This resulted in a significant increase in the average fluorescence across neurons during the active W and WL periods compared to Q periods (*Figure 6f*). The increase in the frequency of WL also correlated with pupil size (*Figure 6g* and *Supplementary file 1 - Table 4*), which reflects the arousal level of the animal (*Busse et al., 2017*).

## Cell-specific encoding of whisking-dependent information in distributed VPM subnetworks

We investigated how neuronal activity was modulated by an important behavioral variable: whether the mouse was whisking or not. We considered neuronal activity both at the single-cell and

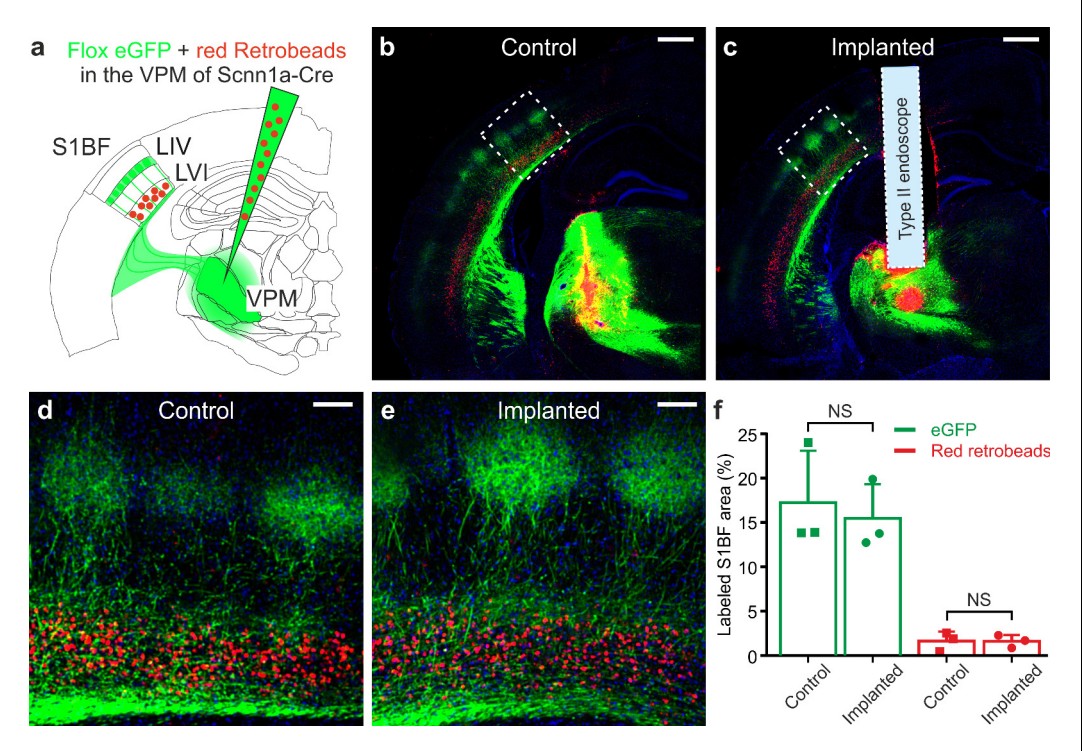

**Figure 5.** Ultrathin microendoscope implantation preserves thalamo-cortical and cortico-thalamic connectivity between S1bf and VPM. (a) Local injection of red retrobeads and AAVs transducing floxed eGFP was performed in the VPM of Scnn1a-Cre mice. (b) Confocal image showing a coronal slice from an injected control animal. Scale bar: 500 μm. (c) Same as (b) but for a mouse implanted with a type II *eFOV*-microendoscope (probe diameter: 500 μm). (d,e) Zoom in of the S1bf region highlighted in (b,c). Scale bar: 100 μm. (f) Percentage of labeled S1bf area with eGFP (green) and retrobeads (red) in control and implanted mice. Points indicate the value of fluorescence from three mice (counted three coronal slices from each animal), column bars indicate average ± sd. One-tailed Mann-Whitney, p=0.20 for eGFP and p=0.50 for red retrobeads, respectively. The online version of this article includes the following source data for figure 5:

**Source data 1.** Percentage of labeled S1bf area with eGFP and retrobeads in control and implanted mice.

population level. We quantified the content of mutual information about whisking state (whether or not the mouse was whisking; shortened to whisking information hereafter) based on the fluorescence signals extracted from individual neurons (*Figure 7a*). We found that many neurons were informative about whisking, but only a fraction were particularly informative (*Figure 7b*). Highly-informative neurons were sparse and could be surrounded by low-information-containing neurons (*Figure 7a*). This indicates that while informative neurons are distributed across the FOV, information is strongly localized and highly cell-specific. Whisking information in individual cells measured with the *eFOV*-microendoscope was constant over the radial distance (*Figure 7c*), with no significant difference in the amount of information in ROI positioned at low radial distance (Dist. low in *Figure 7d*) and at high radial distance (Dist. high in *Figure 7d*). Moreover, adding neurons recorded at higher radial distances to a population decoder improved the amount of extracted whisking information compared to considering only neurons in the center of the FOV (*Figure 7e*). This did not imply that ROIs away from the center carry higher information (information in individual neurons was independent on the cell position within the FOV, *Figure 7c*), but only that information of cells in lateral portions of the FOV summed to the information of more central ROIs in a way that was not purely redundant. Together, these results suggest that the corrected endoscope was effective at capturing information also at the FOV borders. Information of single cells correlated positively with the SNR of their calcium signals (*Figure 7f*), with higher amount of information carried, on average, by cells displaying higher SNR (*Figure 7g*). This suggests that the benefit of *eFOV*-microendoscopes in providing higher SNR, demonstrated earlier, also translates in an ability to extract more information about the circuit's encoding capabilities (*Figure 7e*).

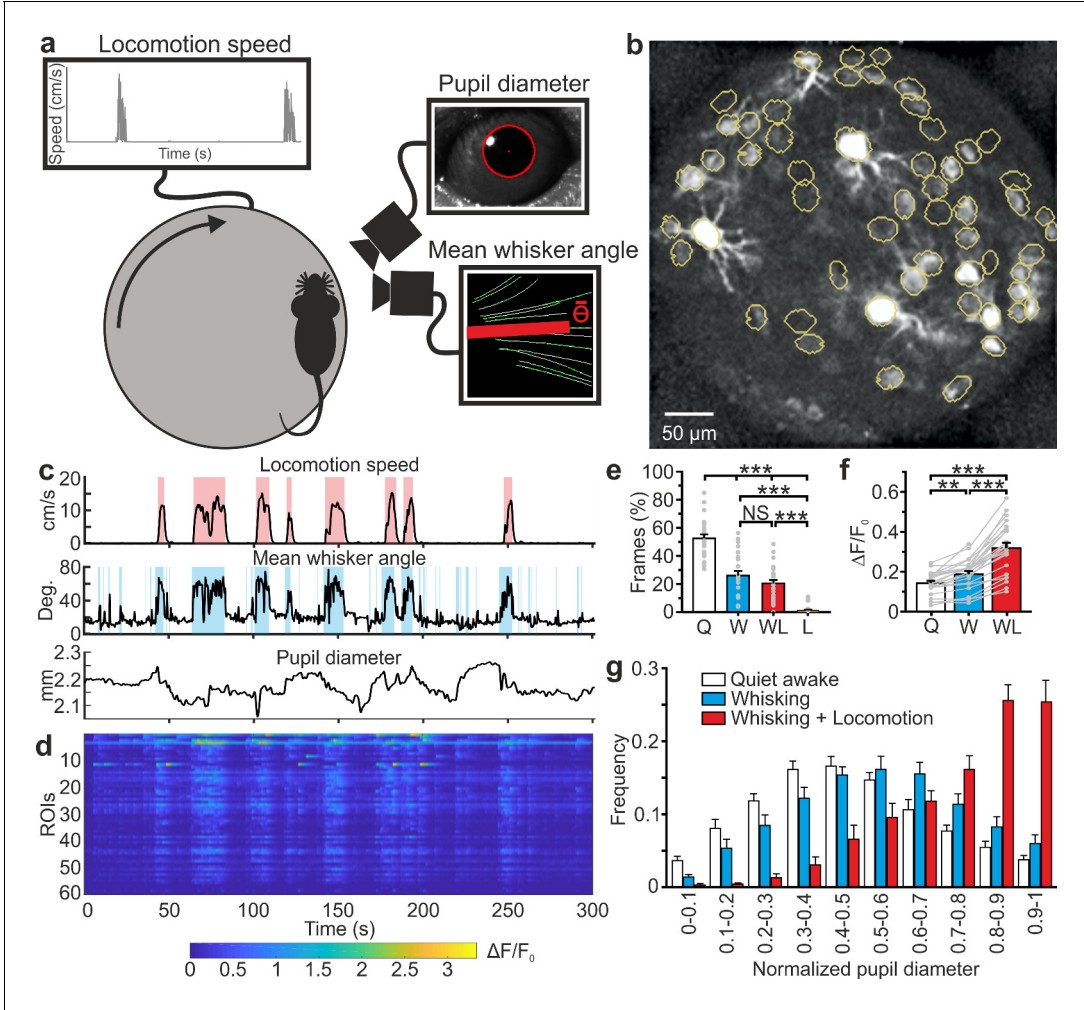

**Figure 6.** High-resolution population dynamics in the VPM of awake mice during locomotion and free whisking. (a) Schematic of the experimental set-up for the recording of locomotion, whisker mean angle, and pupil size in awake head-fixed mice during VPM imaging using type II *eFOV*-microendoscopes. (b) Two-photon image of GCaMP6s labeled VPM neurons in vivo. (c) Representative traces of locomotion (top), whisker mean angle (middle), and pupil diameter (bottom). Red and blue shades indicate periods of locomotion and whishing, respectively (see Materials and methods for definition). (d), $\Delta F/F_0$ over time for all different ROIs in the experiment shown in (c). (e) Percentage of frames spent by the animal in quite wakefulness (Q), whisking (W), locomotion (L), and whisking + locomotion (WL). n = 24 time series from four animals. One-way ANOVA with Bonferroni *post-hoc* correction, p=3E-23. (f) Average $\Delta F/F_0$ across ROIs under the different experimental conditions. n = 24 time series from four animals. One-way ANOVA with Bonferroni *post-hoc* correction, p=2E-16. (g) Distribution of the Q, W, and WL states as a function of pupil diameter. Kolmogorov-Smirnov test for comparison of distributions of Q, W and WL states: p=0.07 for Q vs. W states, p=2E-10 for Q vs. WL states, p=1E-5 for W vs. WL states. For the statistical comparison of Q, W and WL states in each range of pupil diameter a two-way ANOVA with Tukey-Kramer *post-hoc* correction was performed; see *Supplementary file 1 - Table 4*. The error bars represent s.e.m across different *t*-series.

The online version of this article includes the following source data for figure 6:

**Source data 1.** Percentage of frames, $\Delta F/F_0$, and distribution of pupil diameter across behavioral states.

We also considered the redundancy and synergy of whisking information carried by pairs of simultaneously recorded nearby (distance between neurons < 20 μm) neurons. This analysis is important because how pairwise correlations shape the redundancy and synergy of information representation is fundamental to the understanding of population codes (*Runyan et al., 2017*; *Rumyantsev et al., 2020*; *Averbeck et al., 2006*). We found that redundancy and synergy of pairs recorded with the *eFOV*-microendoscopes had, on average, more negative and more positive values in pairs with stronger correlations, respectively (*Figure 7h*). Importantly, and as a consequence of the fact that *eFOV*-microendoscopes avoid the artificial increase of pairwise correlations close to the FOV borders, we did not observe an increase of synergy or redundancy between nearby neurons close to the

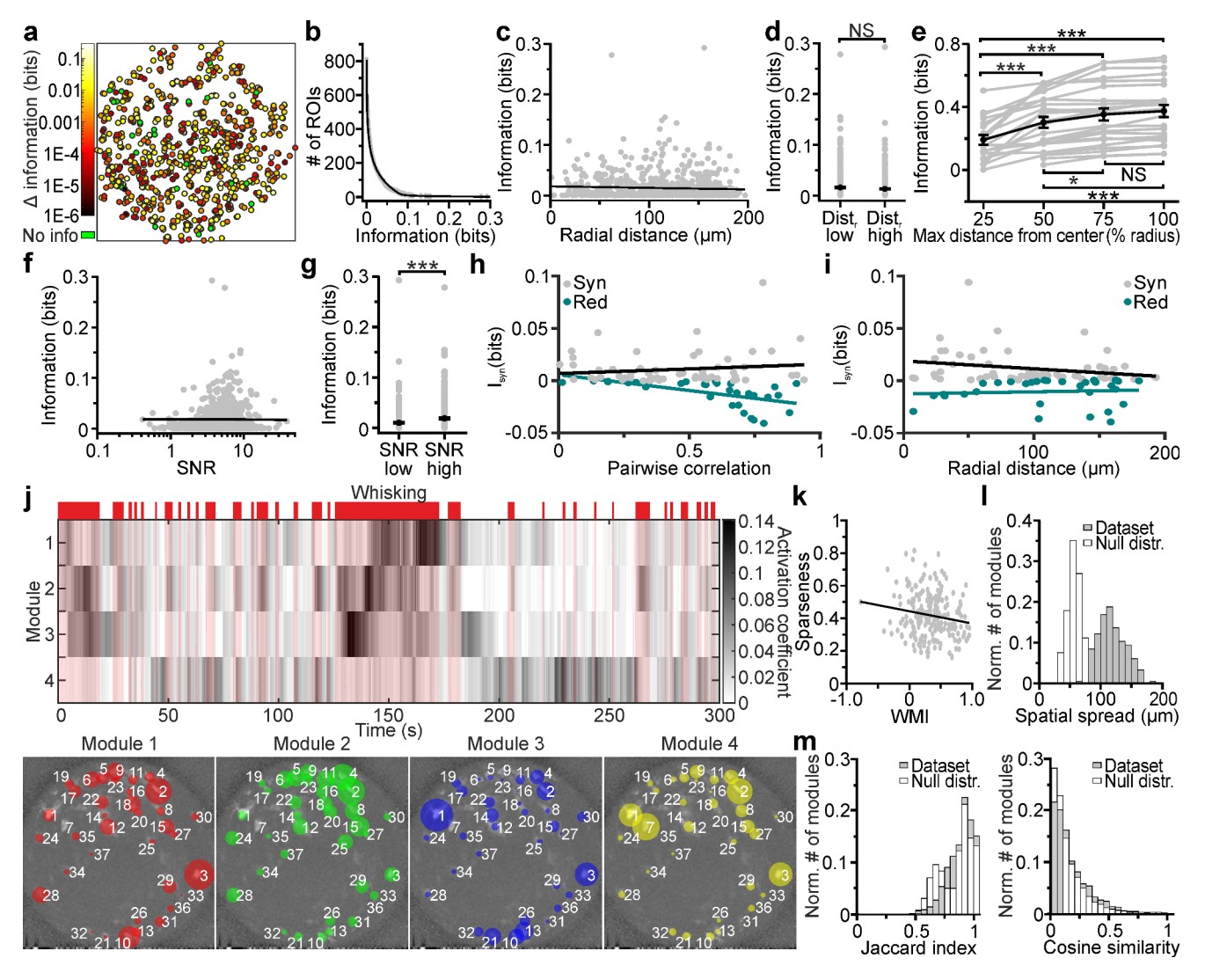

**Figure 7.** Cell-specific encoding of behaviorally-dependent information in distributed VPM subnetworks. (**a**) Spatial map of neurons encoding whisking information. The pseudocolor scale shows significantly informative neurons (see Materials and methods). Data are pooled from 24 *t*-series from four animals. See also *Figure 7—figure supplement 1*. (**b**) Distribution of information content in individual ROIs. The individual ROIs information content distribution was fitted using a double exponential function (R2 = 0.99). n = 808 ROIs from 24 time series. (**c**) Information content of individual neurons vs. their radial distance from the center of the FOV. n = 842 neurons from 24 time series. Linear regression fit: slope = −3E-5, Student's *t*-test p=0.13. Pearson correlation coefficient: −0.053, Student's *t*-test p=0.13. (**d**) Information content for ROIs located at low radial distance (dist$_r$ low) and in lateral portions of the FOV (dist$_r$ high). n = 425 and 417 ROIs from 24 *t*-series for dist$_r$ low and dist$_r$ high, respectively. Mann-Whitney test, p=0.42. (**e**) Information content of neural populations as a function of the distance from the FOV center (% radius) used to include ROIs in the population. One-way ANOVA repeated measurements with Bonferroni *post hoc*, p=1E-18. Data pooled from 24 time series. (**f**) Information content of individual neurons vs. SNR. n = 842 neurons from 24 time series. Linear regression fit: slope = 9E-4, Student's *t*-test p=2.7E-4. Pearson correlation coefficient: 0.125, p=2.7E-4. (**g**) Information content for ROIs with low SNR (SNR low) and high SNR (SNR high). n = 421 and 421 ROIs from 24 *t*-series for dist$_r$ low and dist$_r$ high, respectively. Mann-Whitney test, p=5E-5. (**h,i**) Synergistic (gray) or redundant (dark green) information within pairs of neurons is shown as a function of pairwise correlation (**h**) and as a function of radial distance (**i**). n = 61 and 31 pairs of neurons for synergistic and redundant information content, respectively. Data from 24 time series. Linear regression fit in (**h**): slope = 0.008, permutation test p=0.45 and slope = −0.02, permutation test p=0 for synergistic and redundant information, respectively. Linear regression fit in (**i**): slope = −0.0001, permutation test p=0.007 and slope = 2E-5, permutation test p=0.38 for synergistic and redundant information, respectively. (**j**) Top: representative grayscale matrix showing the activation coefficient across time for 4 NMF modules emerging in a FOV containing 37 neurons. Periods of whisking are shown in red bars and shades. Bottom: ROIs belonging to the four different modules shown in the top panel. Each colored circle represents a ROI belonging to the specified module and its radius is proportional to the ROI weight within that module. The corresponding activation coefficients are presented in the upper panel. (**k**) Module sparseness as a function of the whisking modulation index (WMI). n = 213 modules from 24 time series. Pearson correlation coefficient: −0.164,

*Figure 7 continued on next page*

*Figure 7 continued*

p=0.016. (**l**) Normalized number of modules as a function of their spatial spread for the experimental data (gray) and for a null distribution obtained by randomly shuffling the spatial position of ROIs in the FOV (white). Number of modules: n = 213 and n = 2400 for the 24 *t*-series of the in vivo dataset and for the null distribution, respectively. (**m**) Left: Normalized number of modules as a function of their Jaccard index for pairs of modules identified within the same FOV for experimental data (gray) and for a null distribution (white) obtained by randomly shuffling the spatial position of ROIs within the FOV. Number of modules: n = 1704 and n = 34080 for the 24 *t*-series of the experimental data and for the null distribution, respectively. Right: same as in the left panel for Cosine similarity coefficients.

The online version of this article includes the following source data and figure supplement(s) for figure 7:

**Source data 1.** Information theoretical analysis and non-negative matrix factorization results.
**Figure supplement 1.** Cell-specific encoding of behavioral state-dependent information in distributed VPM subnetworks.

FOV border (*Figure 7i*). This shows that aberration correction helps avoiding the generation of artificially biased estimates of synergy and redundancy near the FOV border.

We finally turned to analyzing the properties of firing at the level of the whole population recorded in the FOV. We applied non-negative matrix factorization (NMF) (*Lee and Seung, 1999*) to identify subpopulations of neurons (modules, *Figure 7j*) characterized by correlated activity. Detected modules were differentially activated in time and were sparsely distributed in space (*Figure 7j*). Moreover, we found that modules could be oppositely modulated by whisking, with the activity of some modules being enhanced and the activity of some other modules being depressed by whisking (*Figure 7j*). We computed the whisking modulation index (WMI, see Materials and methods for definition) and found that the large majority of modules was positively modulated by whisking (WMI >0 for 89.6 ± 0.4% of total modules number, n = 24 FOVs), while the activity of a minority of modules was suppressed during whisking periods (negatively modulated, WMI <0 for 10.4 ± 0.4% of total modules number, n = 24 FOVs). Sparseness of modules appeared to be negatively correlated with the WMI, suggesting that those few modules that were negatively modulated by whisking were also characterized by few, but highly-informative neurons (within the ensemble) (*Figure 7k*). In contrast, modules with high WMI values were less sparse, suggesting similar activity (and information) across most of the neurons belonging to these ensembles. Single modules covered distances of hundreds of μm spanning the whole *eFOV* (*Figure 7l*). We showed that the spatial distances covered by functional modules were higher than distances obtained by chance, using a permutation test (*Figure 7l*). This suggests that corrected probes allow unveiling functional relationship between groups of neurons spanning the whole *eFOV*. Neurons could belong to different ensembles as quantified by the distribution of the values of the Jaccard index among pairs of modules (*Figure 7m*, left). This distribution showed a peak toward the value 1 (the two modules were composed by the same ROIs). However, when ROIs belonged to more than one module they tended to have module-specific weight (i.e. different weights for different modules). In fact, the distribution of values for the cosine similarity, an index which considers the weight of ROIs within a module (see Materials and methods), was shifted toward smaller values compared to the distribution of the Jaccard index values (*Figure 7m*).

## Discussion

Improving optical performances in ultrathin (diameter ≤0.5 mm) microendoscopes with built-in optical elements is a major technical challenge. Since the insertion of the probe irreversibly damages the tissue above the target area, reducing the size of the probe and consequently its invasiveness is of utmost importance when imaging deep brain regions. In this study, we designed, developed, characterized, and successfully validated a new approach to correct aberrations in ultrathin GRIN-based endoscopes using aspheric lenses microfabricated with 3D micro-printing based on TPL (*Liberale et al., 2010*).

Corrective lenses were fabricated on glass coverslips, which were aligned and assembled with the GRIN rod to form an aberration-corrected microendoscope. This optical design resulted in improved axial resolution and extended effective FOV (*Figure 3*), in good agreement with the predictions of the optical simulations (*Figure 2*). The absolute values of the axial PSF computed from optical simulations (*Figure 2—figure supplements 1–2*) were generally smaller in the optical simulations compared to real measurements (*Figure 3* and *Figure 2—figure supplements 1–2*). This may be due to

multiple reasons. First, high-order aberrations were not included in the simulations. Second, in simulations, although the intensity of the excitation PSF was small in lateral portion of the FOV (*Figure 2*), a Gaussian function could still well fit the dim intensity distribution and provide a clear quantification of the PSF dimension. In experimental measurements of fluorescence emitted by subresolved beads, the more degraded PSF in the lateral portions of the FOV would result in low efficacy of the excitation beam in stimulating fluorescence which would result in low SNR fluorescence signals and introduce large variability in the fit. Third, variabilities in some of the experimental parameters (e.g. the distance between the GRIN back end and the focusing objective) were not considered in the simulations.

Aberration correction in GRIN microendoscopes can be achieved using adaptive optics (AO) (*Wang and Ji, 2013*; *Wang and Ji, 2012*; *Bortoletto et al., 2011*; *Lee and Yun, 2011*). For example, using pupil-segmentation methods for AO, diffraction-limited performance across an enlarged FOV was obtained in GRIN-based endoscopes with diameter of 1.4 mm (*Wang and Ji, 2013*; *Wang and Ji, 2012*) and, in principle, this approach could be extended to probes with smaller diameter. Nevertheless, AO through pupil segmentation requires significant modification of the optical setup and the use of an active wavefront modulation system (e.g. a liquid crystal spatial light modulator) which needs the development of *ad-hoc* software control. Moreover, AO through pupil segmentation may limit the temporal resolution of the system, since multiple AO corrective patterns must be applied to obtain an aberration-corrected extended FOV (*Wang and Ji, 2013*). Compared to AO approaches, the technique developed in this study does not necessitate modification of the optical path nor the development of *ad-hoc* computational approaches. Moreover, it is easily coupled to standard two-photon set-ups, and does not introduce limitations in the temporal resolution of the imaging system. A potential alternative to the approach described in this study would be to place a macroscopic optical element of the desired profile in a plane optically conjugated to the objective back aperture along the optical path. This solution could have the advantage of being manufactured using a more standard techniques. However, it would require significant change in the optical set-up in contrast to the built-in correction method that we describe in the present study. Moreover, this macroscopic optical element would have to be changed according to the type of microendoscope used.

Using synthetic calcium data, we demonstrated that the improved optical properties of *eFOV*-microendoscopes directly translate into important advantages for measuring neural population parameters in vivo. Namely, they achieve a higher SNR of calcium signals and a more precise evaluation of pairwise correlations compared to uncorrected GRIN lenses, predictions that were all confirmed experimentally in awake mice (*Figure 4*). Importantly, synthetic calcium data also allowed us to evaluate the impact of correcting optical aberrations on the accuracy in extracting neuronal activity and population codes from calcium imaging data. We found larger correlation of extracted calcium traces with the known ground truth of neuronal spiking activity in *eFOV*-microendoscopes compared to uncorrected probes. Traces extracted from *eFOV*-microendoscopes were less contaminated by neighboring neurons compared to uncorrected probes, in agreement with the higher and more homogeneous spatial resolution of *eFOV*-microendoscopes (*Figure 4*). All these achievements were obtained without increasing the lateral size of the probe, thus minimizing tissue damage in biological applications.

Studying neuronal population codes requires the measurement of neuronal population activity with high-precision, large SNR, and without introducing artificial bias on the activity of individual neurons and the measures of relationship between them, such as pairwise correlations. In particular, pairwise correlations are thought to be fundamental for information coding, signal propagation, and behavior (*Ni et al., 2018*; *Salinas and Sejnowski, 2001*; *Shahidi et al., 2019*; *Panzeri et al., 1999*; *Runyan et al., 2017*). Here, we demonstrate that the homogeneous spatial resolution, which characterized *eFOV*-microendoscopes (*Figure 3*), allowed an unbiased computation of pairwise correlations, a higher correlation of extracted calcium traces with the ground truth neuronal activity, and a smaller contamination of extracted signals by neighboring cells (*Figure 4*). Several studies suggested that even small biases in measuring single cell and pairwise properties, either, for example, in incorrectly measuring the average amount of correlations or the heterogeneity of single cell tuning and of correlation values, may lead to large biases in determining how these populations encode information (*Shamir and Sompolinsky, 2006*; *Panzeri et al., 1999*; *Ecker et al., 2011*). *eFOV*-microendoscopes allow to remove the artifacts and biases introduced by uncorrected GRIN endoscopes in

measuring both individual cell information properties and correlation between each and every pair of neurons. The advantages introduced by *eFOV*-microendoscopes are therefore essential for unraveling the true nature of population codes in deep brain structure.

Corrected endoscopes were characterized by a curved FOV. In the case of type II corrected endoscopes, the change in the z coordinate in the focal plane was up to ~75 μm (*Figure 3*). This z value was smaller for all other corrected endoscope types (*Figure 3*). The observed field curvature of corrected endoscopes may impact imaging in brain regions characterized by strong axially organized anatomy (e.g. the pyramidal layer of the hippocampus), but would not significantly affect imaging in regions with homogeneous cell density within the z range described above (<75 μm for type II corrected microendoscopes).

We used the unique features of the *eFOV*-microendoscopes to study how highly correlated activity is mapped in the VPM thalamic nucleus of awake mice with unprecedented combination of high spatial resolution across the FOV and minimal invasiveness (*Figures 5–7*). The VPM is a primary somatosensory thalamic nucleus which relays sensory information to the S1bf (*Sherman, 2012*). However, VPM also receives strong cortical innervation which deeply affects VPM activity (*Crandall et al., 2015*; *Mease et al., 2014*; *Temereanca and Simons, 2004*). We first showed that the small cross-section of the *eFOV*-microendoscopes developed here preserved thalamocortical and corticothalamic connectivity (*Figure 5*), a fundamental prerequisite for VPM physiology and a hardly achievable task with larger cross-section GRIN lenses or with chronic windows (*Figure 3—figure supplement 7e*). We then imaged GCaMP6s-expressing VPM neurons while monitoring locomotion, whisker movement, and pupil diameter (*Figure 6*). We found cell-specific encoding of whisking information in distributed functional VPM subnetworks. Most individual neurons encoded significant amount of whisking information generating distributed networks of informative neurons in the VPM (*Figure 7*). However, the amount of encoded information was highly cell-specific, with high-information-containing neurons being sparsely distributed in space and surrounded by low-information-containing cells. Sparse distribution of information content has been similarly observed in other brain areas (*Runyan et al., 2017*; *Ince et al., 2013*). At the population level, we observed the presence of whisking modulated functional ensembles of neurons, which were oppositely modulated by whisking. Some ensembles displayed enhanced activity upon whisking, while some other ensembles showed suppressed activity by whisking. Interestingly, single neurons could belong to multiple functional ensembles, but their weight within one ensemble was ensemble-specific (*Figure 7*). Overall, the application of *eFOV*-microendoscopes revealed the complexity and cellular specificity of the encoding of correlated behavioral state-dependent information in a primary thalamic sensory nucleus.

One important area of future development for *eFOV*-microendoscopes will be to determine whether the approach we described in this study could be used to correct optical aberrations in GRIN lenses different from the ones described here (e.g. longer GRIN rods). A second area of interest will be to develop corrective lenses or compound corrective lenses for 3D imaging in larger volumes. This should be possible given that, in the present study, we designed the corrective lenses to maximize aberration correction only in the focal plane of the endoscope and in optical simulations we found that the Strehl ratio was >0.8 in a 1.7–3.7 times larger volume in corrected compared to uncorrected endoscopes (*Figure 2—figure supplement 3*). Finally, new fabrication materials (*Weber et al., 2020*) may allow to develop solutions for effective chromatic aberrations (*Figure 2—figure supplement 4*) in two-photon multi-wavelength applications.

In summary, we developed a new methodology to correct for aberrations in ultrathin microendoscopes using miniaturized aspheric lenses fabricated with 3D printing based on TPL. This method is flexible and can be applied to the GRIN rods of different diameters and lengths that are required to access the numerous deep regions of the mammalian brain. Corrected endoscopes showed improved axial resolution and up to nine folds extended effective FOV, allowing high- resolution population imaging with minimal invasiveness. Importantly, we demonstrated that *eFOV*-microendoscopes enable more precise extraction of population codes from two-photon imaging recordings. Although *eFOV*-microendoscopes have been primarily applied for functional imaging in this study, we expect that their use can be extended to other applications. For example, *eFOV*-microendoscopes could be combined with optical systems for two-photon holographic optogenetic manipulations (*Packer et al., 2012*; *Rickgauer and Tank, 2009*; *Papagiakoumou et al., 2010*) and for simultaneous functional imaging and optogenetic perturbation (*Packer et al., 2015*;

*Rickgauer et al., 2014*; *Carrillo-Reid et al., 2016*; *Forli et al., 2018*; *Marshel et al., 2019*) using a diffractive optical element to provide patterned illumination of neurons but also to correct for z-defocus with an appropriate lens function. Moreover, besides its applications in the neuroscience field, *eFOV*-microendoscopes could be used in a large variety of optical applications requiring minimally invasive probes, ranging from cellular imaging (*Kim et al., 2012*; *Ghosh et al., 2011*) to tissue diagnostic (*Huland et al., 2012*; *Huland et al., 2014*). Importantly, applications of ultrathin *eFOV*-microendoscopes to other fields of research will be greatly facilitated by the built-in aberration correction method that we developed. This provides a unique degree of flexibility that allows using ready-to-use miniaturized endoscopic probes in a large variety of existing optical systems with no modification of the optical path.

# Materials and methods

## Key resources table

| Reagent type (species) or resource | Designation | Source or reference | Identifiers | Additional information |
|---|---|---|---|---|
| Strain, strain background (*M. musculus*) | C57BL/6J | Charles River | RRID:IMSR_JAX:000664 | |
| Genetic reagent (*M. musculus*) | B6;C3-Tg(Scnn1a-cre)3Aibs/J | The Jackson Laboratory | RRID:IMSR_JAX:009613 | |
| Recombinant DNA reagent | pAAV.Syn.Flex.GCaMP6s.WPRE.SV40 | Penn Vector Core | RRID:Addgene_100845; Addgene viral prep # 100845-AAV1 | *Chen et al., 2013* |
| Recombinant DNA reagent | pGP-AAV-syn-FLEX-jGCaMP7f-WPRE | Addgene | RRID:Addgene_104492; Addgene viral prep # 104492-AAV1 | *Dana et al., 2016* |
| Recombinant DNA reagent | AAV pCAG-FLEX-EGFP-WPRE | Penn Vector Core | RRID:Addgene_51502; Addgene viral prep # 51502-AAV1 | *Oh et al., 2014* |
| Recombinant DNA reagent | AAV.CaMKII0.4.Cre.SV40 | Penn Vector Core | RRID:Addgene_105558; Addgene viral prep # 105558-AAV1 | |
| Commercial assay or kit | Kwik-Cast | World Precision Instruments | Cat# KWIK-CAST | |
| Commercial assay or kit | Sylgard Silicone Elastomer | Dow Inc | Cat# Sylgard 164 | |
| Commercial assay or kit | Norland Optical Adhesive 63 | Norland | Cat# NOA 63 | |
| Commercial assay or kit | GRIN lens | Grintech | Cat# NEM-050-25-10-860-S | |
| Commercial assay or kit | GRIN lens | Grintech | Cat# NEM-050-43-00-810-S-1.0p | |
| Commercial assay or kit | GRIN lens | Grintech | Cat# GT-IFRL-035-cus-50-NC | |
| Commercial assay or kit | GRIN lens | Grintech | Cat# NEM-035-16air-10–810 S-1.0p | |
| Chemical compound, drug | bisBenzimide H 33342 trihydrochloride (Hoechst) | Sigma-Aldrich | Cat# B2261; CAS: 23491-52-3 | |
| Chemical compound, drug | Red Retrobeads | LumaFluor Inc | Red Retrobeads | |
| Software, algorithm | Zemax OpticStudio 15 | Zemax | https://www.zemax.com/products/opticstudio | |
| Software, algorithm | MATLAB R2017a | Mathworks | RRID:SCR_001622; https://it.mathworks.com/products/matlab.html | |

*Continued on next page*

*Continued*

| Reagent type (species) or resource | Designation | Source or reference | Identifiers | Additional information |
|---|---|---|---|---|
| Software, algorithm | GraphPad PRISM | GraphPad PRISM | RRID:SCR_002798; https://www.graphpad.com/ | |
| Software, algorithm | ImageJ/Fiji | Fiji | RRID:SCR_002285; http://fiji.sc/ | |
| Software, algorithm | NoRMCorre | *Pnevmatikakis and Giovannucci, 2017* | https://github.com/flatironinstitute/NoRMCorre | |
| Software, algorithm | CalmAn | *Giovannucci et al., 2019* | https://github.com/flatironinstitute/CalmAn-MATLAB | |
| Software, algorithm | Population Spike Train Factorization Toolbox for Matlab Version 1.0 | *Onken et al., 2016* | https://stommac.eu/index.php/code | |
| Software, algorithm | LIBSVM | *Chang and Lin, 2011* | https://www.csie.ntu.edu.tw/~cjlin/libsvm/ | |
| Software, algorithm | Information Breakdown ToolBox | *Magri et al., 2009* | N/A | |
| Software, algorithm | Software used in this paper for generation of artificial time series | https://github.com/moni90/eFOV_microendoscopes_sim | | *Figure 4a–h, Figure 4—figure supplement 1a–c* and *Figure 4—figure supplement 2a–c* |
| Software, algorithm | Software to compute recall, precision, and F1 score | *Soltanian-Zadeh et al., 2019* | https://github.com/soltanianzadeh/STNeuroNet | |
| Other | Basler ace camera | Basler AG | Cat# acA800-510um | |
| Other | Optical encoder | Broadcom | AEDB-9140-A13 | |
| Other | Zortrax M200 3D printer | Zortrax | M200 | |
| Other | Z-ULTRAT 3D printer filament | Zortrax | Z-ULTRAT | |
| Other | Arduino Uno | Arduino | Arduino Uno | |

## Animal models

Experimental procedures involving animals have been approved by the Istituto Italiano di Tecnologia Animal Health Regulatory Committee, by the National Council on Animal Care of the Italian Ministry of Health (authorization # 1134/2015-PR, # 689/2018-PR) and carried out according to the National legislation (D.Lgs. 26/2014) and to the legislation of the European Communities Council Directive (European Directive 2010/63/EU). Experiments were performed on adult (8–14 week old) mice. C57BL/6J mice (otherwise called C57, Charles River #000664, Calco, IT) were used in *Figure 3—figure supplement 6*. Data reported in *Figures 4–7* and *Figure 3—figure supplement 7* were obtained from B6;C3-Tg(Scnn1a-cre)3Aibs/J (JAX #009613, Jackson Laboratory, Bar Harbor, USA) mice crossed with C57 mice (otherwise called Scnn1a-Cre). Both male and female animals were used in this study. Animals were housed in individually ventilated cages under a 12 hr light:dark cycle. Access to food and water was ad libitum. The number of animals used for each experimental set of data is specified in the text or in the corresponding Figure legend.

## Methods details

### Design and simulation of corrective lenses and of *eFOV*-microendoscopes

Simulations were run with OpticStudio15 (Zemax, Kirkland, WA) to define the profile of the aspheric corrective lens to be integrated in the aberration-corrected microendoscopes, with the aim to achieve: (i) a full-width half maximum (FWHM) lateral resolution <1 μm at the center of the FOV; (ii) a FWHM axial resolution below <10 μm; (iii) a working distance between 150 μm and 220 μm into living brain tissue. The wavelength used for simulations was λ = 920 nm. The surface profile of corrective aspheric lenses was described in *Optic Studio Manual, 2017*:

$$Z(r) = \frac{cr^2}{1 + \sqrt{1 - (1+k)c^2r^2}} + \sum_n \alpha_n r^{2n} \tag{1}$$

Since GRIN lenses have intrinsic spherical aberration, the optimization for the shape of the corrective lenses started with the profile of a Schmidt corrector plate (*Born and Wolf, 1999*) as initial guess; the parameters $c$, $k$, $\alpha_n$ (with n = 1–8) in *Equation (1)* were then automatically varied in order to maximize the Strehl ratio (*Dorband et al., 2012*) over the largest possible area of the FOV (*Supplementary file 1 - Table 1*). A fine manual tuning of the parameters was performed for final optimization. Ray-trace simulations were performed considering the material used in lens replica (i.e. NOA63). Simulated two-photon PSFs (*Figure 2—figure supplements 1–2*) were determined by 3D sampling the squared calculated Strehl ratio. Intensity profiles derived from sections in the x, y, z directions of the simulated two-photon PSFs were fitted with Gaussian curves and analyzed as for experimental PSFs (see below).

To evaluate the effect of corrective lenses on the 3D image space (*Figure 2—figure supplement 3*), the Strehl ratio was calculated for different defocused working distances, which were simulated by changing the spacing between the microscope objective and the microendoscope back end. Different radial distances were simulated on the same fields used in *Figure 2*. To evaluate the axial chromatic aberrations (*Figure 2—figure supplement 4*), the Strehl ratio was calculated for different wavelengths. For those wavelengths, the defocused working distance which maximizes the Strehl ratio, was also determined.

## Corrective lens manufacturing and microendoscope assembly

The optimized aspheric lens structure obtained with simulations was exported into a 3D mesh processing software (MeshLab, ISTI-CNR, Pisa, IT) and converted into a point cloud dataset fitting the lens surface (with ~300 nm distance among first neighborhood points). Two-photon polymerization with a custom set-up (*Liberale et al., 2010*) including a dry semi-apochromatic microscope objective (LUCPlanFLN 60x, NA 0.7, Olympus Corp., Tokyo, JP) and a near infrared pulsed laser beam (duration: 100 fs; repetition rate: 80 MHz; wavelength: 780 nm; FemtoFiber pro NIR, Toptica Photonics, Graefelfing, DE) was used for the fabrication of the corrective lenses. A drop of resin (4,4'-Bis(diethylamino)benzophenone photoinitiator mixed with a diacrylate monomer), sealed between two coverslips, was moved by a piezo-controlled stage (model P-563.3CD, PI GmbH, Karlsruhe, DE) with respect to the fixed laser beam focus, according to the 3D coordinates of the previously determined point cloud with precision of 20 nm. Output laser power was ~15 mW at the sample. Once the surface was polymerized, the lens was dipped for ~2 min in methanol followed by ~1 min immersion in isopropyl alcohol and finally exposed to UV light ($\lambda$ = 365 nm; 3 Joule / cm$^2$[*Salinas and Sejnowski, 2001*]) to fully polymerize the bulk of the structure.

For fast generation of multiple lens replicas, a molding (*Schaap and Bellouard, 2013*) technique was used. To this end, polydimethylsiloxane (PDMS, Sylgard 164, 10:1 A:B, Dow Corning, Auburn, MI) was casted onto the lens and hardened by heat cure in a circulating oven at 80°C for approximately 30 min. The resulting bulked structure of solid PDMS was then used as negative mold. A drop of a UV-curable optically-clear adhesive with low fluorescent emissivity (NOA63, Norland Products Inc, Cranbury, NJ) was deposited on the negative mold, pressured against a coverslip (diameter: 5 mm) of appropriate thickness (thickness: 100 or 200 μm depending on the *eFOV*-microendoscope type, *Figure 1*) and hardened by UV exposure. One side of the UV-curable adhesive was in contact with the mold, the other side was instead attached to the coverslip. After UV curing, by gently pulling the glass coverslip away, the lens made of UV-curable adhesive detached easily from the PDMS mold, while remaining firmly attached to the coverslip. The coverglass with the lens attached was then glued onto a metal ring. The yield for 3D printed lenses and lens replica was ~100%. One end of the appropriate GRIN rod (NEM-050-25-10-860-S, type I; NEM-050-43-00-810-S-1.0p, type II; GT-IFRL-035-cus-50-NC, type III; NEM-035-16air-10–810 S-1.0p, type IV, Grintech GmbH, Jena, DE) was attached perpendicularly to the other surface of the coverslip using NOA63. Alignment of the corrective lens and the GRIN rod was performed under visual guidance using an opto-mechanical stage, custom-built using the following components (*Figure 3—figure supplement 2*): camera (DCC1645C, Thorlabs, Newton, NJ), fine z control (SM1Z, Thorlabs, Newton, NJ), coarse z control (L200/M, Thorlabs, Newton, NJ), xyz control (MAX313D/M, Thorlabs, Newton,

NJ), high power UV LED (M375L3, Thorlabs, Newton, NJ), long pass dichroic mirror (FF409-Di02, Semrock, Rochester, NY), tube lens (AC254-150-A, Thorlabs, Newton, NJ), objective (UPlanFLN 4×, 0.13NA, Olympus, Milan, IT), xy control (CXY1, Thorlabs, Newton, NJ), custom GRIN rod holder, and fiber optic holder (HCS004, Thorlabs, Newton, NJ). An additional and removable coverglass or a silicone cap (Kwik-Cast Sealant, World Precision Instruments, Friedberg, DE) was glued on the top of every support ring to keep the polymeric corrective lens clean and to protect it from mechanical damage.

## Optical characterization of *eFOV*-microendoscopes

Optical characterization of *eFOV*-microendoscopes was carried out with a two-photon laser-scanning microscope equipped with a wavelength-tunable, ultrashort-pulsed, mode-locked Ti:Sapphire laser source (Ultra II Chameleon, pulse duration: 160 fs; repetition rate: 80 MHz; wavelength: 920 nm; Coherent Inc, Santa Clara, CA) and a commercial Prairie Ultima IV scanhead (Bruker Corporation, Milan, IT). For all measurements, the wavelength was set at 920 nm. The optomechanical assembly used for the *eFOV*-microendoscope characterization is shown in *Figure 3—figure supplement 2c*. The coupling objective was EC Epiplan-Neofluar 20x, 0.5NA (Zeiss, Oberkochen, DE). The z control (SM1Z) and xy control (CXY2) were purchased from Thorlabs (Newton, NJ). Spatial resolution of each microendoscope was evaluated using subresolved spherical fluorescent beads (diameter: 100 nm, Polyscience, Warrington, PA), following a previous spatial calibration using a custom fluorescent ruler (Motic, Xiamen, CN). The same ruler was used to evaluate the distortion of the FOV. To visualize the curvature of the imaging field, thin (thickness: 300 nm) fluorescent slices (*Antonini et al., 2014*) were used. Fluorescent samples were deposited on a microscope slide and imaged through the endoscope assembly aligned to the microscope objective, with or without the corrective microlens above the coverslip, using the coupling apparatus described in *Figure 3—figure supplement 2*. Imaging was performed with the distal end of the GRIN rod immersed in a droplet of water placed on the slide. We observed no appreciable damage on the lens over imaging sessions. In considering this, please note that the distance between the focal plane of the microscope objective and the endoscope assembly was ~100 μm and it was fixed for all measurements. Given the imaging field curvature of endoscopes, for both the ruler and the thin fluorescent slices (planar samples), the acquisition of z-series of images (512 pixels x 512 pixels, with 1 μm axial step) was performed.

## Viral injections and microendoscope implantation

Adeno-associated viruses (AAVs) AAV1.Syn.flex.GCaMP6s.WPRE.SV40, AAV1.CAG.Flex.eGFP. WPRE.bGH, AAV1.CaMKII0.4.Cre.SV40 were purchased from the University of Pennsylvania Viral Vector Core. AAV1.Syn.flex.GCaMP7f.WPRE.SV40 was purchased from Addgene (Teddington, UK) Animals were anesthetized with isoflurane (2% in 1 L/min $O_2$), placed into a stereotaxic apparatus (Stoelting Co, Wood Dale, IL) and maintained on a warm platform at 37°C. The depth of anesthesia was assessed by monitoring respiration rate, heartbeat, eyelid reflex, vibrissae movements, reactions to tail and toe pinching. 2% lidocaine solution was injected under the skin before surgical incision. A small hole was drilled through the skull and 0.5–1 μl (30–50 nl/min, UltraMicroPump UMP3, WPI, Sarasota, FL) of AAVs containing solution was injected at stereotaxic coordinates: 1.4 mm posterior to bregma (P), 1 mm lateral to the sagittal sinus (L), and 1 mm deep (D) to target the hippocampal CA1 region; 1.7 mm P, 1.6 mm L, and 3 mm D to target the VPM. Co-injection of AAV1.Syn.flex. GCaMP6s.WPRE.SV40 and AAV1.CaMKII0.4.Cre.SV40 (1:1) was performed to express GCaMP6s in hippocampus CA1 pyramidal cells of C57 mice (*Figure 3—figure supplement 6*). Injection of AAV1. Syn.flex.GCaMP6s.WPRE.SV40 (1:4 in saline solution) (*Figures 6–7* and *Figure 3—figure supplement 7*) or AAV1.Syn.flex.GCaMP7f.WPRE.SV40 (1:4 in saline solution) (*Figure 4*) in the Scnn1a-Cre mice was performed to express GCaMP6/7 in the VPM. Following virus injection a craniotomy (~600×600 μm² or ~400×400 μm² depending on the endoscope size) was performed over the neocortex at stereotaxic coordinates: 1.8 mm P and 1.5 mm L to image the hippocampus; 2.3 mm P and 2 mm L to reach the VPM. A thin column of tissue was suctioned with a glass cannula (ID, 300 μm and OD, 500 μm; Vitrotubs, Vitrocom Inc, Mounting Lakes, NJ) and the microendoscope was slowly inserted in the cannula track, using a custom holder, down to the depth of interest and secured by acrylic adhesive and dental cement to the skull. If necessary, metal spacers (thickness:~100 μm) were glued on the flat coverslip surface to obtain the desired protrusion distance of the GRIN rod. For

experiments in awake animals (*Figures 4*, *6* and *7* and *Figure 3—figure supplement 7*), a custom metal head plate was sealed on the skull using dental cement to assure stable head fixation during two-photon imaging. An intraperitoneal injection of antibiotic (BAYTRIL,Bayer, DE) and dexamethasone (MSD Animal Health, Milan, IT) was performed to prevent infection and inflammation. Animals were then positioned under a heat lamp and monitored until recovery.

To evaluate thalamocortical (TC) and corticothalamic (CT) anatomical connectivity (*Figure 5*) after implantation of microendoscopes, Scnn1a-Cre mice were injected, as described above, at 100 nl/min (UltraMicroPump UMP3, WPI, Sarasota, FL) with AAV1.CAG.Flex.eGFP.WPRE.bGH virus (1:2 in saline solution) and red retrobeads (1:8 in saline solution, Fluorescent Latex Microspheres, Luma-Fluor Inc, Durham, NC) in VPM after tissue suctioning and in the absence of tissue suctioning. The total injected volume was 250 nl.

## Functional imaging with *eFOV*-microendoscopes in vivo

For experiments in anesthetized conditions (*Figure 3—figure supplement 6*), three to five weeks after injection, mice were anesthetized with urethane (16.5%, 1.65 g*kg$^{-1}$) and placed into a stereotaxic apparatus to proceed with imaging. Body temperature was measured with a rectal probe and kept at 37°C with a heating pad. Depth of anesthesia was assured by monitoring respiration rate, eyelid reflex, vibrissae movements, and reactions to pinching the tail and toe. In some experiments, oxygen saturation was controlled by a pulseoxymeter (MouseOx, Starr Life Sciences Corp., Oakmont, PA).

For experiments in behaving mice (*Figures 4*, *6* and *7* and *Figure 3—figure supplement 7*), imaging was performed two to four weeks after the endoscope implant, following 7–10 days of habituation, in which mice were placed daily on the set-up, each day for a longer time duration, up to 45 min (*Gentet et al., 2010*). Mice were allowed to run spontaneously on the wheel. During experiments, recording sessions were up to five 5 min long (frame rate typically ~3 Hz) and they were interleaved by 5 min in which no imaging was performed. We did not collect fluorescent signals at frame rates higher than 4 Hz. This was because we aimed at imaging the largest possible FOV and our experimental set-up was equipped with regular galvanometric mirrors. For scanning imaging of GCaMP6/7-expressing neurons, the same microscope set-up used for the optical characterization of *eFOV*-microendoscopes was used and GCaMP6/7 fluorescence was excited at 920 nm (laser power: 28–90 mW).

## Measurement of whisker angle, pupil size, and locomotion

In *Figures 6–7*, whisker movements in the whisker pad contralateral to the recording site were imaged with a high-speed,Basler acA800 (Basler, Ahrensburg, DE; acquisition rate: 150 Hz) through a 45° tilted mirror placed below the whiskers. Illumination was provided by an array of infrared LEDs (emission wavelength: 800 nm) fixed to the microscope objective and aligned to the whiskers and the mirror. Imaging of the contralateral eye was performed during each experimental session with a Basler acA800 (Basler, Ahrensburg, DE), coupled with a long pass filter (HP 900, Thorlabs, Newton, NJ) at 80 Hz acquisition rate. Illumination light was provided by the pulsed laser sourced which was used to perform two-photon microendoscopic imaging (λ = 920 nm). Locomotor activity was measured with an optical encoder (AEDB-9140-A13, Broadcom, San Jose, CA) mounted under the wheel.

## Immunohistochemistry

Deeply anesthetized animals were transcardially perfused with 0.01 M PBS (pH 7.4) followed by 4% paraformaldehyde. Brains were post-fixed for 6 hr, cryoprotected with 30% sucrose solution in 0.1 M PBS, and serially cut in coronal sections (thickness: 40–50 μm) using a HM 450 Sliding Microtome (Thermo Fisher). Sections were counterstained with Hoechst (1:300, Sigma–Aldrich, Milan, IT), mounted, and coverslipped with a DABCO [1,4-diazobicyclo-(2,2,2)octane]-based antifade mounting medium. Fluorescence images were acquired with a Leica SP5 inverted confocal microscope (Leica Microsystems, Milan, IT).

For the evaluation of the anatomical connections between VPM and S1bf, mice were perfused after 10 days from the injection and GRIN lens implantation. 50 μm thick coronal brain slices were cut, counterstained with Hoechst (1:300, Sigma–Aldrich, Milan, IT), and mounted with an Antifade

Mounting Medium (Vectashield, Burlingame, CA). Confocal images were acquired with a Nikon Eclipse scope (Nikon, Milan, IT).

## Simulations

### Geometrical considerations

In *Figure 4a–h*, neurons were simulated as spheres with an average radius $r\_mean$ which was estimated from recorded data ($r\_mean$ = 7.95 ± 2.33 µm, mean ± sd). Some variability was introduced in the neurons size, sampling it from a normal distribution with mean $r\_mean$ and standard deviation $r\_sigma$ = 1.31 (within the measured sd). At the center of each neuron a nuclear region was added, such that the spherical shell surrounding the nucleus had a width of variable size ($r\_shell$ randomly sampled from a normal distribution with mean = 4 µm and sd = 1 µm). The nuclear region did not express GCaMP6s and the fluorescence signal could be collected only from the spherical shell surrounding the nucleus. Simulated neurons were randomly placed in a volume of size 500 × 500×80 µm³ with no overlap between neurons up to the point that the volume was filled with cells or that neural density reached the value 83,100 ± 7,900 cells/mm³ (*Keller et al., 2018*). The resolution of the spatial volume was 0.5 µm/pixel in the x and y direction, 1 µm/pixel in the z direction.

### Neural activity

Neural spiking activity was simulated as the sum of Poisson processes. Each neuron was assigned with a mean spiking rate ($rho$ = 0.4, arbitrary selected), following a binary synchronicity matrix with value one for neurons with common inputs (*common inputs probability* = 0.8, arbitrary selected). The activity of each neuron was the sum of an independent Poisson process and as many common Poisson processes as the neurons with shared variability due to common inputs. The spiking rate of the summed Poisson processes was the mean spiking rate for that neuron. We assigned to randomly selected groups of neurons a shared common input generated as a Poisson process. This resulted in average pairwise correlations in the ground truth calcium activity larger than zero (average correlation 0.084 ± 0.001, n = 5833) and independent from the radial distance. Calcium activity and fluorescence traces were then generated using the equations in [*Friedrich et al., 2017*]. An autoregressive model of order 1 (with parameter $\gamma$ = 0.7) was selected to convolve spike trains into calcium activity. A model with supralinearities and Hill saturation was used to convert calcium activity into fluorescence intensity (model parameters: baseline = 0; single event amplitude = 1500; Hill saturation coefficient = 1; dissociation constant = 1; noise variance = 0.05).

### Generation of fluorescence time series

The size and the resolution of the simulated FOV were set to 500 × 500 µm² and 2.5 µm/pixel, respectively. The resolution was adjusted according to the changes in the magnification factor (estimated from experimental data *Figure 3—figure supplement 4*), obtaining a non-uniform resolution in the FOV.

To generate the synthetic *t*-series, we used the measurements experimentally obtained from corrected and uncorrected microendoscopes (*Figure 3*). For corrected microendoscopes, the synthetic imaging focal surface was a spherical shell with curvature radius estimated from the measurements of fluorescent films (curvature radius: 400 µm) and the excitation volume was an ellipsoid resembling the aberration-corrected and experimentally measured PSF. For uncorrected microendoscopes, the synthetic imaging surfaces were two spherical shells with curvature radius estimated from the data (curvature radius: 265 µm and 2000 µm, respectively) and the excitation volume was an ellipsoid resembling the aberration-uncorrected and experimentally measured PSF.

Excitation volumes were scanned along the imaging focal surface (or surfaces for uncorrected microendoscopes), such that their axial direction was always orthogonal to the imaging focal surface (s). All the voxels falling within the excitation volumes contributed to the signal of the corresponding pixel in the FOV, resulting in one of the following possible three conditions:

- If the pixel was in the edge of the FOV (radial distance >250 µm), its signal was randomly sampled from a normal distribution, with mean and standard deviation estimated from experimental data. For this and the following conditions, we selected the best fitting distribution for the signal mean through log-likelihood maximization across four alternative models: a normal, a gamma, a log-normal distribution, and a Gaussian mixture model. Dark noise mean was best

fitted by a Gaussian mixture model (component 1: proportion = 0.37; mean = 137.48; sd = 48.96; component 2: proportion = 0.63; mean = 126.83; sd = 5.02, Log-likelihoods: Normal = $-3.3E5$, Gamma = $-3.0E5$, Log-normal = $-2.9E5$, Gaussian mixture model = $-2.6E5$), while the standard deviation of the dark noise depended on the dark noise mean in a linear way ($p_0$ = $-175.39$, $p_1$ = 1.57). The simulated dark noise was generated with the mean randomly sampled from the Gaussian Mixture Modeling (GMM) distribution and the standard deviation linearly dependent from the mean.

- If the pixel was in the central part of the FOV (radial distance ≤250 µm) but no neurons were within the excitation volume, the pixel signal was randomly sampled from a normal distribution with mean and standard deviation estimated from experimental data. The mean intensity of pixels that were neither in the edges nor belonging to ROIs were fitted using a lognormal distribution (mean = 5.43, sd = 0.36. Log-likelihoods: Normal = $-1.46E6$, Gamma = $-1.39E6$, Log-normal = $-1.37E6$, Gaussian mixture model = $-1.39E6$) and the best linear fit between the squared root of the mean intensity and the intensity sd was computed ($p_0$ = $-162.55$, $p_1$ = 18.28). Simulated noise in the FOV was generated as Gaussian noise with mean randomly sampled from the lognormal distribution and sd linearly dependent from the squared root of the mean.

- If the pixel was in the central part of the FOV (radial distance ≤250 µm) and at least one neuron was in the excitation volume(s), each voxel in the excitation volume(s) was assigned either Gaussian noise (estimated as in the previous condition. Log-likelihoods: Normal = $-2.70E5$, Gamma = $-2.60E5$, Log-normal = $-2.58E5$, Gaussian mixture model = $-2.59E5$) in case no neurons were in that voxel, or the fluorescence intensity of the neuron sampled by that voxel. In case a neuron was contained in a voxel, Gaussian noise was also added to the neuron signal. The mean of the added Gaussian noise was zero, while the sd was proportional to the square root of the mean intensity of the voxel, with the coefficients estimated from a linear fit between the square root of the mean intensity and the intensity standard deviation of pixels assigned to ROIs in experimental data ($p_0$ = $-132.44$, $p_1$ = 16.94). The activity of all the voxels falling within the excitation volume(s) was then averaged to obtain the pixel's fluorescence intensity. The intensity of each pixel signal was finally modulated as a function of the radial position within the FOV, accordingly to the optical characterization of corrected and uncorrected microendoscopes using the radial intensity obtained imaging the subresolved fluorescent layer (*Figure 3*).

In simulations, the imaging rate of the *t*-series was set to 5 Hz.

## Segmentation of simulated time series

We segmented simulated time series using two approaches: an automated procedure that we developed to resemble manual segmentation, and a standard automated approach (CaImAn [*Giovannucci et al., 2019*]). The automated procedure that we developed for the segmentation of synthetic *t*-series was based on the ground truth spatial distribution of neurons in the FOV. We first associated with each imaged neuron its spatial footprint, which consists of all the pixels collecting signal from that neuron. This segmentation would be the ideal segmentation. However, for experimental data, the ground truth is not available neither to users segmenting the FOV nor to automated segmentation algorithms. For this reason, we modified the ideal segmentation to obtain a more realistic situation. We reasoned that very small ROIs are not likely to be detected and therefore we removed all the ROIs with few pixels. We tried different thresholds for the minimum number of pixels composing a ROI (n = 5,10,15 pixels). We observed that this parameter did not have an effect on the results of the comparison between corrected and uncorrected microendoscopes and therefore we set it to 5. We then reasoned that overlapping ROIs could be distinguished only if their overlap was not perfect and therefore we merged ROIs with high overlap. The overlap between two ROIs was defined as the fraction between overlapping pixels and total number of pixels of the smallest ROI. We merged ROIs with overlap larger than 70, 80% and 90%. This parameter had no effect on the results of the comparison between corrected and uncorrected microendoscopes and we set it to 80%. Ultimately, we considered that ROIs whose fluorescence signal had low SNR could not be discriminated from noise. We reasoned that a single event was sufficient to segment a ROI and we defined a peak-SNR as the ratio between the maximum peak in the fluorescence trace and the baseline noise (defined as the sd of those parts of the fluorescence trace with signal lower than the 25% of the fluorescence distribution). We considered in the final segmentation only ROIs whose signal peak-SNR was higher than a threshold of 5, 10, 15, 20, 25, and 30 (*Figure 4*).

For the CaImAn segmentation, we tested different values of the SNR parameter (0.25, 0.5, 1, 1.5, 2), that represented a lower SNR threshold for a ROI to be kept in the final segmentation.

## Statistics and analysis

### Statistics

Values are expressed as mean ±sem, unless otherwise stated; the number of samples (n) and p values are reported in the Figure legends or in the text. No statistical methods were used to pre-determine sample size. All recordings with no technical issues were included in the analysis and blinding was not used in this study. Statistical analysis was performed with MATLAB software (Mathworks, Natick, MA) and GraphPad Prism software (GraphPad Software, San Diego, CA). A Kolmogorov-Smirnov test was run on each experimental sample to test for normality and to test the equality of the distributions in *Figures 6g* and *7m*. The significance threshold was always set at 0.05. When comparing two paired populations of data, paired Student's *t*-test or paired Wilcoxon signed-rank test (*Figure 4c*, *Figure 4—figure supplement 1d,e*) were used to calculate statistical significance in case of normal and non-normal distribution, respectively. Unpaired Student's *t*-test (*Figure 4k*, *Figure 4—figure supplement 2b,d*) and Mann-Whitney test (*Figures 4e,l* and *7d,g*) were used for unpaired comparisons of normally and non-normally distributed data, respectively. One-way analysis of variance (ANOVA) with Bonferroni post-hoc correction was used to compare the dependence of multiple classes from a single factor (*Figures 6e,f* and *7e*). Two-way ANOVA with Bonferroni post-hoc correction was used to compare the dependence of multiple classes from two factors (*Figure 6g*). Two-way ANOVA with the interaction factor and with Bonferroni post-hoc correction was used in *Figure 4d* and *Figure 4—figure supplement 2a*. To fit linear data, linear regression was used. The significance of linear regression coefficients being different from zero was assessed using a permutation test, where we built a null distribution of linear regression coefficients by first destroying the relationship between the variable of interest (n = 5000 repetitions) and then fitting a linear regression model to each repetition of the permuted data (*Figures 4g,m* and *7h,i*, and *Figure 4—figure supplement 2c,e*). Pearson correlation coefficients were used to test the dependence between variables (*Figure 7c,f,k*). The significance of the Pearson correlation coefficients was assessed using a Student's *t*-test. All tests were two-sided, unless otherwise stated. Information theoretical analyses, NMF module identification, module characterization, and SVM classification were performed on MATLAB software (Mathworks, Natick, MA) using available toolboxes (*Magri et al., 2009*) or custom written codes.

### Analysis of confocal images

Three mice were unilaterally injected with AAV-eGFP and red retrobeads after tissue aspiration and then implanted with the endoscope. Three mice were injected with AAV-GFP and red retrobeads without tissue aspiration and they were not implanted (controls). Three confocal images were acquired from fixed slices for each hemisphere at different focal planes (minimal distance between planes: 20 µm). Images in the red and green acquisition channels were blurred with a Gaussian filter (sigma = 2 µm) and binarized with a triangle thresholding method. S1bf was manually identified using anatomical cues from Hoechst labeling in each sample. To quantify the amount of preserved TC and CT connections within a given area of S1bf, we computed the fraction of pixels showing suprathreshold pixel intensity out of the total of pixel of the chosen area. A single value for each sample, obtained by averaging between confocal images of different FOVs, was used to run the one-tailed Mann-Whitney test for the different acquisition channels (*Figure 5*).

### Analysis of field distortion and calibration of the pixel size

A regular fluorescent grid spanning the FOV was imaged in order to evaluate the distortion in the FOV. The number of pixels necessary to span 10 µm in the x and y direction was measured as a function of the distance from the FOV center. A magnification factor which varied along the radial directions was evaluated by computing the ratio between the measured number of pixels in the distorted (microscope objective coupled with GRIN-lens-based microendoscope) and undistorted (microscope objective alone) conditions. The estimated magnification factor (from x and y directions) was fitted using a quadratic curve (corrected: $p_0 = 0.76$, $p_1 = -6.24E-04$, $p_2 = 1.95E-05$, norm of residuals = 1.78; Uncorrected: $p_0 = 0.73$, $p_1 = -2.91E-04$, $p_2 = 8.11e-06$, norm of residuals = 0.24). The

magnification factor was used to correctly calibrate experimental measurements in *Figures 3, 4, 6* and *7*, *Figure 3—figure supplement 3*, *Figure 4—figure supplements 1–2*, and *Figure 7—figure supplement 1*.

To measure the PSF as a function of the radial position within the FOV, z-stacks of subresolved fluorescent beads (diameter: 100 nm) were taken at different distances from the optical axis. Intensity profiles obtained from sections in the x, y, z directions of the PSFs were fitted with Gaussian curves and their FWHM was defined as x, y, and z resolution, respectively. Lateral resolution was calculated as the average of x and y resolution. Axial resolution coincided with the z resolution. When, due to aberrations in the lateral portion of the FOV, the intensity profile in the z direction was better fitted with a sum of two Gaussian curves instead of a single one, the axial resolution was defined as the axial distance between the two peaks of the best fitting curves. For each group of measurements at a specific distance from the optical axis, outliers were identified using the Rout method (Graph-Pad Software, San Diego, CA) and excluded from the data. Mean and standard deviation of resolutions were plotted against radial distance (*Figure 3*). Data were fitted with a symmetric quartic function to respect the cylindrical geometry of the optical system and the maximal FOV radial extent was determined as the radial distance at which the axial resolution fitting curve crossed a 10 μm threshold. PSF measurements were conducted using at least three eFOV-microendoscope for each type.

## Analysis of fluorescence *t*-series

Experiments in VPM were analyzed with a customized graphical user interface (GUI) in MATLAB (version R2017a; Mathworks, Natick, MA). The GUI enabled the loading of data saved from the microscope acquisition software, included the motion correction algorithm (NoRMCorre) described in *Pnevmatikakis and Giovannucci, 2017*, facilitated the manual segmentation of the FOV, and allowed the deconvolution of neural activity from the recorded fluorescent dynamics. ROIs were drawn by visualizing single frames or temporal projections of the FOV and, within the selected regions, only those pixels necessary to maximize the peak-signal-to-noise ratio (SNR) of the mean intensity were selected. Peak-SNR was defined as:

$$peakSNR = \frac{\max\limits_{t \in [t'-\delta t, t'+\delta t]} f(t) - avg(f_{baseline})}{std(f_{baseline})}$$

where $f_{baseline}$ is the portion of intensity trace lower than the 25[th] percentile of the intensity distribution and t' is the peak time instant. After segmentation, the GUI performed the deconvolution of normalized calcium activity from the fluorescence extracted from ROIs, by using the algorithm provided in *Giovannucci et al., 2019*; *Pnevmatikakis et al., 2016*. The algorithm was based on the fit of the fluorescence activity with an autoregressive model. We used models of order 1 if the acquisition rate was low (<2 Hz), otherwise order 2. At the end of the pre-processing, a structure containing all the extracted information (acquisition parameters, ROIs spatial footprints, ROIs fluorescence activity, deconvolved activity, and normalized calcium activity) was saved and used for the subsequent analyses.

## Analysis of synthetic and experimental time series
### Quantification of ROIs number

For synthetic data in *Figure 4* and *Figure 4—figure supplement 2*, the number of segmented ROIs was computed as a function of the probe type (uncorrected or corrected microendoscopes), of the peak-SNR value used in the segmentation procedure, and of the interaction between these two factor.

For experimental data in *Figures 4, 6* and *7*, background intensity was not always uniform in the endoscopes FOV and in some regions no ROIs could be detected. In order to discount this factor, the count of the ROIs was normalized to the brighter part of the FOV, obtaining a measurement of the ROIs density (number of ROIs divided by the total bright area). To detect the dark background regions, the edges of the FOV (where mostly dark noise was collected) were used as intensity threshold. All the parts of the FOV with mean intensity lower than 85[th] percentile of the threshold were discarded for the normalization of the ROIs count. The remaining part of the FOV was considered as

bright area and used for the ROIs density analyses. For the benchmarking of manual and automatic segmentation in *Figure 4—figure supplement 1*, we computed recall, precision, and F1 score using the code provided in *Soltanian-Zadeh et al., 2019*.

## SNR of calcium activity
For both synthetic and experimental data, the SNR was defined as:

$$SNR = \frac{power(Cdf)}{power(Fraw - Cdf)} = \frac{\sum_i (Cdf(t_i))^2}{\sum_i (Fraw(t_i) - Cdf(t_i))^2},$$

where $F_{raw}$ and $Cdf$ are the z-scored raw fluorescence intensity and deconvoluted calcium activity, respectively. SNR was evaluated to measure the quality of the extracted ROIs signal.

## Correlation with ground truth activity
For synthetic data, we computed the correlation between the calcium activity of each segmented ROI and the ground truth calcium activity of neurons contributing to that ROI. In case more neurons were merged during the automated segmentation that we developed, we sorted the merged neurons for decreasing correlation with the corresponding ROI. We defined 'source neuron' the neuron with highest correlation with the ROI (*Figure 4h* left) and considered the correlation with the other merged neurons (only the second highest correlation is shown) as a measure of signals' contamination between nearby neurons (*Figure 4h* right).

## Pairwise correlation
For each pair of nearby ROIs (distance between ROIs centers < 20 µm), we computed pairwise correlation of the extracted calcium activity as a function of the radial distance of the ROIs pair. We defined the radial distance of each pair as the distance between the FOV's center and the center of the segment connecting the two ROIs' centers. To measure changes in pairwise correlations exclusively caused by changes in imaging resolution, we normalized the pairwise correlations of nearby ROIs by subtracting the value of pairwise correlations between distant ROIs (distance >60 µm) placed at the same radial distance. We then fitted the normalized pairwise correlation as a function of radial distance using linear regression.

## Analysis of behavioral parameters
Videos of whisker movements were binarized with the Ridge Detection plugin of ImageJ/Fiji in order to individuate pixels corresponding to whiskers. Videos were then processed in MATLAB (Mathworks, Natick, MA) to extract the whisker mean angle. To this aim, all whiskers were fitted with straight lines and for each frame the mean angle of all the lines was calculated with respect to the horizontal direction of the FOV. Once the mean angle of the imaged whiskers was calculated for each frame, this signal was processed with a moving standard deviation over a 400 ms window and a Gaussian filter over a 50 ms window. Whisking and no whisking periods were identified by binarizing the mean whisker angle with a temporal and amplitude threshold. While the temporal threshold was fixed at 200 ms, the amplitude threshold was extracted by manually identifying whisking periods in ~1/10th of the full-length video and using this manual classification to find the best amplitude threshold with a ROC analysis. Temporal gaps between whisking periods shorter than 0.5 s were considered whisking periods, and linear interpolation was used to obtain the whiskers mean angle in frames in which less than four whiskers were detected.

For the analysis of pupil diameter, movies were analyzed with MATLAB (Mathworks, Natick, MA). Each frame was thresholded with the Otsu's method and the region corresponding to the pupil was approximated with an ellipse. The length of the major axis of the ellipse was considered the pupil diameter. Linear interpolation was used for frames in which the pupil was not properly detected.

Detection of locomotion periods was performed using a threshold criterion on the wheel speed (*Pakan et al., 2016*). The wheel speed signal was downsampled at 40 Hz and an instant was considered to be part of a locomotion epoch if it met the following conditions: (i) instantaneous speed >1 cm/s; (ii) low-pass filtered speed (short pass filter at 0.25 Hz) >1 cm/s; (iii) average speed over 2 s

windows >0.1 cm/s. Temporal gaps between locomotion periods shorter than 0.5 s were considered periods of locomotion.

Four behavioral states were defined in *Figure 6*: (i) quiet (Q) when neither locomotion nor whisking was observed, (ii) whisking (W), when whisking but no locomotion was observed; (iii) locomotion (L), when locomotion but not whisking was detected; (iv) whisking and locomotion (WL), when both locomotion and whisking were detected. L epochs were extremely rare (1.45 ± 0.75% of total acquisition time, mean ± sem) and were not considered in the analysis. For the SVM analysis and for the NMF analysis (*Figures 6* and *7*), we considered just two states: quiet (Q) and active (A), with A being the union of W and WL.

## Analysis of calcium signals across states

To compare the amplitude of calcium activity across behavioral states, the deconvolved activity of each ROI was averaged in each of the three states (Q, W and WL).

To measure whether calcium activity was further modulated by arousal, we discretized the pupil size in ten bins and measured the distribution of the average calcium activity for each behavioral state, separately.

## Information theoretic analysis

For information theoretical analyses, we used the MATLAB toolbox provided in *Magri et al., 2009*. To compute whisking information encoded in the calcium signal of single ROIs, we computed the amount of mutual information (*Quian Quiroga and Panzeri, 2009*) that the calcium signals carried about whether the animal was in a quiet (Q) or active (A) state. For each state, $n_T$ time points, where $n_T$ is the number of time points spent in the less frequent state, were randomly sampled without replacement. For each ROI then, the calcium activity in the selected time points was used to compute the information carried by that ROI regarding the state of the animal (the details were as follows: direct method, quadratic extrapolation for bias correction, number of response bins = 2). The amount of information was considered significantly different from 0 only if the real information value was larger than the 95th percentile of the distribution of n = 500 shuffled information values (obtained after randomly shuffling the whisking state label). This procedure was repeated n = 100 times, randomly sampling different time points and the reported information values were computed as the average information encoded across different iterations.

We sorted ROIs according to their information content and fitted the distribution of information content of individual ROIs using a double exponential function, where the information carried by the *i*-th ROI was given by:

$$I_{ROI_i} = a * \exp(-b * i) + c * \exp(-d * i)$$

We used the R (*Salinas and Sejnowski, 2001*) coefficient to assess the goodness of the fit. We computed the Pearson correlation coefficient to check for dependence between the information carried by individual ROIs and the ROIs radial distance (or the ROIs SNR). To test whether ROIs at different radial distances (or with different SNR) carried a different amount of information, we split the ROIs in two groups. ROIs with low radial distance (or SNR) where those ROIs whose radial distance (or SNR) was lower than the median of the radial distances (or SNR) distribution. ROIs with high radial distance (or SNR) where those ROIs whose radial distance (or SNR) was higher (or equal) than the median of the radial distances (or SNR) distribution.

To compute information carried by pairs of neurons, we considered only pairs of nearby neurons (distance between neurons < 20 μm). We computed the amount of synergistic information carried by the neuronal pair as a function of the pairwise correlation between neurons and as a function of the pair's radial distance. Data in *Figure 7h* were permutation subtracted. Specifically, information values were computed for n = 500 random permutations of the whisking variable label of the data, which destroyed the relationship between the whisking variable and the neuronal response. We then subtracted from the value of information obtained for real data the mean of the information obtained across permutations. We used the same permutations to compute a null distribution for synergy and redundancy values. n = 43 out of n = 61 pairs (70.4%) had significant synergy. n = 22 out of n = 31 pairs (70.1%) had significant redundancy. We fitted the synergistic information using a linear function for synergistic and redundant pairs, separately. We computed the linear fit using both

all pairs (shown in *Figure 7h*) and only pairs showing significant synergy/redundancy (pairwise correlations: Synergy, slope = 0.009, permutation test p=0.2; Redundancy, slope = 0.05, permutation test p=0. Radial distance: Synergy, slope = $-1E-4$, permutation test p=0.002; Redundancy, slope = 7.1E-6, permutation test p=0.46). Results were not affected by this choice.

To compute information about the whisking state from a large population of neurons, we first decoded the whisking state from the single-trial population activity, and then we expressed the decoding performance as mutual information contained in the confusion matrix as in Equation (11) of *Quian Quiroga and Panzeri, 2009*. We performed single-trial population decoding as follows. A Gaussian kernel SVM was trained to classify the animal state (Q or A) by observing population activity (*Chang and Lin, 2011*). For each state, $n_T$ time points were randomly sampled (without replacement) and split into two equal and balanced sets. One set was used as training set for the SVM, and on this data, a ten-fold cross validation was performed over a fixed grid for the SVM hyperparameters. The performance of the SVM was then tested using the test set. This procedure was repeated n = 100 times by randomly sampling different time points. The reported classification accuracy was the average information encoded across different iterations. To check whether an increase in the number of imaged ROIs led to a better classification of whisking state (*Figure 7e*), we computed information for neuronal population of gradually increasing size. At first, we considered only the ROIs in the central portion of the FOV (distance from FOV center < ¼ of FOV radius) and then the other ROIs (distance steps = ¼ of FOV radius) were incrementally added for the training and testing phase of the SVM.

## Non-negative matrix factorization (NMF)

To compute NMF (*Figure 7*), two states (Q and A) were considered. For each state, $n_T$ time points were randomly sampled (without replacement) and split into two equal and balanced sets (a training set and a test set). The number of NMF modules was selected based on the ability of a linear discriminant analysis (LDA) classifier trained on the reduced data to predict the presence or absence of whisking (*Onken et al., 2016*). The dimensionality of the training set and the test set were at first reduced to k, with k ranging from one to the number of ROIs of the dataset. Then, for each factorization, the LDA classifier was trained on the training set to predict the behavioral state variable, and its performance was tested on the test set. The final dimension for the NMF was selected as the number of modules at which the first elbow in the performance plot (performance increase <0.4%) was found. Then, the dimensionality of the entire dataset was reduced by computing the NMF with the selected number of modules. For each module in the obtained factorization, the following quantities were computed:

- Sparseness: $sparseness = \frac{\sqrt{\sum_j \left(w_j^{(i)}\right)^2}}{\sum_j w_j^{(i)}}$, where $w_j^{(i)}$ denotes the contribution of the j$^{th}$ ROI to the i$^{th}$ module. Sparseness values close to 1 indicate that few ROIs contribute heavily to the module, while sparseness values close to 0 indicate that the contribution of ROIs to the modules is more homogeneous.

- Whisking modulation index (WMI): $WMI = \frac{mean(act_A) - mean(act_Q)}{mean(act_A) + mean(act_Q)}$, where $act_A$ and $act_Q$ denote the activation coefficients in each behavioral state. WMI > 0 indicates that the module's activity is increased during A state, while WMI < 0 indicates that the module's activity is reduced during A state.

- Spatial spread: we defined as spatial spread the shortest path that connected the ten ROIs with highest weights (or all the ROIs in case a module was composed by fewer ROIs).

To compare pairs of modules, we computed the following similarity measures (*Figure 7m*):

- The Jaccard index is the fraction between the number of ROIs belonging to both modules and the total number of ROIs composing the two modules (without repetitions). It ranges between 0 and 1 and assumes value 0 if two modules do not share common ROIs, value one if two modules are composed by exactly the same ROIs. The Jaccard index does not take into account the weights of ROIs in the modules.

- Cosine similarity = $\frac{\sum w_i * v_i}{\sqrt{\sum w_i^2} * \sqrt{\sum v_i^2}}$, where $w_i$ and $v_i$ represent the weight of the $i$-th ROI in each of the two modules. As for the Jaccard index, cosine similarity ranges between 0 and 1 and it takes value 0 if two modules do not share common ROIs. Contrarily to the Jaccard index, cosine similarity takes into account also the weights of the ROIs in the modules and it is 1 only if two ROIs are composed by exactly the same ROIs with equal weights.

For both the Jaccard index and Cosine similarity, we computed null distributions. Specifically, for each NMF factorization we reassigned ROIs randomly within each module by shuffling their weights. We did not consider modules composed by single ROIs.

## Data and software availability

The datasets shown in *Figures 4*, *6* and *7* and corresponding figure supplements are available at: https://doi.org/10.17632/wm6c5wzs4c.1.

The software used in this paper to generate artificial *t*-series is available at: https://github.com/moni90/eFOV_microendoscopes_sim (*Antonini, 2020*; copy archived at https://github.com/elifesciences-publications/eFOV_microendoscopes_sim).

# Acknowledgements

We thank M Dal Maschio for discussion at an initial stage of the project, F Nespoli for preliminary analysis, and B Sabatini and A Begue for critical reading an early version of this manuscript. We thank DS Kim and the GENIE project for the constructs Addgene viral prep # 100845-AAV1 and #104492-AAV1, H Zeng for the construct Addgene viral prep # 51502-AAV1, and JM Wilson for the construct Addgene viral prep # 105558-AAV1. This work was supported by an IIT interdisciplinary grant and in part by ERC (NEURO-PATTERNS), NIH Brain Initiative (U01 NS090576, U19 NS107464, R01NS109961), FP7 (DESIRE), MIUR FIRB (RBAP11 × 42L), and Flag-Era JTC Human Brain Project (SLOW-DYN). CL acknowledges support from KAUST under baseline funding BAS/1/1064-01-01.

# Additional information

## Funding

| Funder | Grant reference number | Author |
| --- | --- | --- |
| European Research Council | NEURO-PATTERNS | Tommaso Fellin |
| National Institutes of Health | BRAIN Initiative NS090576 | Tommaso Fellin |
| National Institutes of Health | BRAIN Initiative NS107464 | Stefano Panzeri Tommaso Fellin |
| National Institutes of Health | BRAIN Initiative NS109961 | Stefano Panzeri |
| Seventh Framework Programme | DESIRE | Tommaso Fellin |
| FIRB | RBAP11X42L | Tommaso Fellin |
| Flag-Era JTC Human Brain Project | SLOW-DYN | Stefano Panzeri Tommaso Fellin |
| IIT interdisciplinary grant | | Carlo Liberale Tommaso Fellin |
| King Abdullah University of Science and Technology | BAS/1/1064-01-01 | Carlo Liberale |

The funders had no role in study design, data collection and interpretation, or the decision to submit the work for publication.

## Author contributions

Andrea Antonini, Data curation, Formal analysis, Validation, Investigation, Visualization, Methodology, Writing - original draft; Andrea Sattin, Monica Moroni, Data curation, Software, Formal analysis,

Validation, Investigation, Visualization, Writing - original draft; Serena Bovetti, Data curation, Software, Formal analysis, Validation, Investigation, Visualization, Methodology, Writing - original draft; Claudio Moretti, Data curation, Software, Formal analysis, Validation, Investigation, Visualization, Methodology; Francesca Succol, Validation, Investigation, Methodology; Angelo Forli, Formal analysis, Supervision, Validation; Dania Vecchia, Formal analysis, Visualization, Writing - original draft; Vijayakumar P Rajamanickam, Validation, Methodology; Andrea Bertoncini, Methodology; Stefano Panzeri, Formal analysis, Supervision, Methodology, Writing - original draft, Writing - review and editing; Carlo Liberale, Conceptualization, Supervision, Funding acquisition, Methodology, Writing - original draft, Project administration, Writing - review and editing; Tommaso Fellin, Conceptualization, Resources, Supervision, Funding acquisition, Writing - original draft, Project administration, Writing - review and editing

### Author ORCIDs

Monica Moroni  https://orcid.org/0000-0003-1852-7217
Dania Vecchia  https://orcid.org/0000-0002-6091-538X
Stefano Panzeri  http://orcid.org/0000-0003-1700-8909
Carlo Liberale  https://orcid.org/0000-0002-5653-199X
Tommaso Fellin  https://orcid.org/0000-0003-2718-7533

### Ethics

Animal experimentation: Experimental procedures involving animals have been approved by the Istituto Italiano di Tecnologia Animal Health Regulatory Committee, by the National Council on Animal Care of the Italian Ministry of Health (authorization # 1134/2015-PR, # 689/2018-PR) and carried out according to the National legislation (D.Lgs. 26/2014) and to the legislation of the European Communities Council Directive (European Directive 2010/63/EU).

### Decision letter and Author response

Decision letter https://doi.org/10.7554/eLife.58882.sa1
Author response https://doi.org/10.7554/eLife.58882.sa2

## Additional files

### Supplementary files

• Supplementary file 1. Characteristics of eFOV-microendoscopes and their application in awake mice. Supplementary Table 1: Parameters for the fabrication of corrective lenses. Coefficients used in *Equation (1)* (see Materials and methods) for the aspherical corrective lenses used in type I-IV *eFOV*-microendoscopes. Supplementary Table 2: Simulated focal length in uncorrected and corrected microendoscopes. Supplementary Table 3: Spatial resolution and effective FOV of *eFOV*-microendoscopic probes. Values are reported as average ± sem. For statistical comparison of uncorrected (uncor.) vs. corrected (cor.) microendoscopes, Student's *t*-test was used. Supplementary Table 4: Statistical comparisons of behavior state distributions as a function of pupil diameter. For the statistical comparison of Q, W, and WL state distributions in each range of pupil diameter, a two-way ANOVA with Tukey-Kramer *post-hoc* correction was performed.

• Transparent reporting form

### Data availability

The datasets shown in Figures 1-7 and corresponding figure supplements are available at: https://doi.org/10.17632/wm6c5wzs4c.2. The software used in this paper to generate artificial t-series is available at: https://github.com/moni90/eFOV_microendoscopes_sim (copy archived at https://github.com/elifesciences-publications/eFOV_microendoscopes_sim) Numerical data for graphs represented in figures 3-7, figure 2-figure supplement 2-4, figure 3-figure supplement 4, figure 4-figure supplement 1-2 are provided as source data.

The following dataset was generated:

| Author(s) | Year | Dataset title | Dataset URL | Database and Identifier |
|---|---|---|---|---|
| Antonini A, Sattin A, Moroni M, Bovetti S, Moretti C, Succol F, Forli A, Vecchia D, Rajamanickam VP, Bertoncini A, Panzeri S, Liberale C, Fellin T | 2020 | Dataset of Extended field-of-view ultrathin microendoscopes for high-resolution two-photon imaging with minimal invasiveness | https://doi.org/10.17632/wm6c5wzs4c.2 | Mendeley Data, 10.17632/wm6c5wzs4c.2 |

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
