## [Decision Letter]

**Acceptance summary:**

In this revised manuscript, the authors demonstrated an extended field-of-view (eFOV), thin (<500μm) microendoscopes by adding aberration correcting microlenses fabricated by TPL. The performance of the microlens corrected eFOV endoscopes was evaluated with simulations on synthetic calcium data and in vivo activity imaging. In addition, the eFOV microendoscopes were used to investigate VPM activity correlated with locomotion, whisker movement, and pupil diameter. Cell-specific encoding in VPM during these behaviors is further scrutinized with statistical and machine learning methods. The presentation of the methods and the results are clear. The methods are useful and practical for increasing the FOV of ultrathin microendoscopes, removing one of the current limitations of small GRIN lens applications. The authors significantly improved the manuscript based on the reviewer's comments.

**Decision letter after peer review:**

Thank you for submitting your article "Extended field-of-view ultrathin microendoscopes for high-resolution two-photon imaging with minimal invasiveness" for consideration by *eLife*. Your article has been reviewed by three peer reviewers, and the evaluation has been overseen by a Reviewing Editor and Andrew King as the Senior Editor. The following individuals involved in review of your submission have agreed to reveal their identity: Darcy S Peterka (Reviewer #2); Kaspar Podgorski (Reviewer #3).

The reviewers have discussed the reviews with one another and the Reviewing Editor has drafted this decision to help you prepare a revised submission.

Your manuscript describes the development and use of microfabricated corrector lenses for improving GRIN lens imaging. These correction lenses increase the usable field-of-view (FOV) of a given GRIN lens, and improves the resolution. The approach was validated in a variety of samples. While the reviewers agree that the work will be potentially high impact, they feel that revisions with additional analyses would significantly improve the paper.

Reviewer #1:

In this manuscript, the authors demonstrated an extended field-of-view (eFOV), thin (<500μm) microendoscopes by adding aberration correcting microlenses fabricated bt TPL. Performance of the microlens corrected eFOV endoscopes was demonstrated with simulations on synthetic calcium data and in vivo activity imaging. In addition, the eFOV microendoscopes were used to investigate VPM activity correlated with locomotion, whisker movement, and pupil diameter. Cell-specific encoding in VPM during these behaviors are further scrutinized with statistical and machine learning methods. The presentation of the methods and the results are clear. The methods are useful and reasonably practical for increasing the FOV of ultrathin microendoscopes, which is a major limitation for small GRIN lenses. Overall, I recommend publication of the paper in *eLife*. Listed below are a number of concerns that we hope the authors can address in their revision to further improve the manuscript.

1) For the design and fabrication of the eFOV-microendoscope:

a) The authors presented four types of GRIN lenses (Type I-IV). Four different models of the GRIN lens are shown in subsection “Corrective lens manufacturing and microendoscope assembly”. It will probably help if the authors can explicitly indicate the Types with the model number.

b) It is claimed that the experimental result is similar to ray-trace simulation as shown in Figure 2. However, there is no comparison to support the claim. The high-order coefficient in the gradient profile in GRIN lens is usually not included in the optical model used for ray-trace. High-order aberration will thus not be reflected in the simulation. It would be important to know the difference between the simulation result and the experimental result. For example, the authors can add experimental data in Figure 2 or add simulation data in Figure 3.

c) For all the experiments, is it performed with the 2-photon polymerized lens with resin or with lens replica using a molded UV-curable adhesive? Is the aberration correction performance different from these two fabrication methods? Which material is the ray-tracing simulation based on?

d) What is the yield of the lens fabricated? How is the cured UV-curable adhesive detached from the PDMS mold? During imaging sessions, was there any damage on the lens?

e) How does the correction microlens affect the focal length of the GRIN lens. In other words, what is the allowable change in working distance before and after the correction?

f) There appears to be a much larger field curvature with the corrected GRIN lens. It would be helpful to discuss the impact of this on functional imaging.

g) Instead of AO with an active element (e.g., SLM), one could also put a fixed lens with the desired curvature in the place of the SLM. Compared to the microlens approach, one needs to modify the excitation path somewhat, but the lens can be fabricated using standard techniques and may be more accessible to most research labs. It might be informative to the readers to comment on this alternative approach.

2) For the data taken with the eFOV-microendoscope, there is ~2x improvement in FOV for type III lens (Figure 3G). However, in Figure 3—figure supplement 5C, the images before and after correction for type III lens is almost the same.

3) For the data simulation and analysis:

a) Analysis was shown to correlate whisking, locomotion and cell-specific encoding. Figure 6C and 6D show that locomotion is usually correlated with a higher ΔF/F. Could this be a result of the motion artifact during locomotion from the head-constraint mouse?

b) To simulate synthetic calcium data, different models (Gaussian Mixture, lognormal, etc.) are used to obtain the pixel intensity value as described in subsection “Generation of fluorescence time series”. It is mentioned that the model is obtained by fitting the experimental data. What criteria (e.g. comparing likelihood) is used to choose those specific models? What other distribution is tested to choose the best-fit model?

c) As mentioned in subsection “Analysis of field distortion and calibration of the pixel size”, the author used a gaussian profile to define axial resolution. It is probably better to fit with a Lorentzian function I(z)=〖(1/(1+〖(z/zR)〗^2 ))〗^2 for 2-photon excitation, where zR is the Rayleigh range of the gaussian focus. Not sure how large the difference will be between these two fits.

d) Figure 4D. It is not clear to me why the ROI number is smaller for the corrected lens than the uncorrected when the peak SNR value is small (e.g., less than 10). Can the authors add some explanations?

e) The authors claim that "pairwise correlation increased as a function of the radial distance of the pair in uncorrected probes compared to corrected ones." Figure 4M does not seem to show noticeable increase as a function of the radial distance for the uncorrected lens (left panel). Rather, it seems to show a noticeable decrease as a function of the radial distance for the corrected lens (right panel). Therefore, the statement does not seem to be supported by Figure 4M. This is confusing, and some explanation is needed.

4) There is a typo in the first sentence of subsection “Fabrication of eFOV-microendoscopes”. I believe it is (Figure 3—figure supplement 1C) instead of (Figure 1C).

Reviewer #2:

In this paper, the authors describe the creation and use of microfabricated corrector lenses for improving GRIN lens imaging. GRIN lenses are one of a few technologies (and the only one that is "mature") that allow for high resolution optical imaging in brains at depths beyond ~1.5 mm. These correction lenses a) increase the usable field-of-view (FOV) of a given GRIN lens, and b) improve the fidelity of signals extracted from regions-of-interest (ROIs), by decreasing the point spread function (PSF) volume. Both are important for high fidelity recording of neuronal populations. The paper is well written, and I find their demonstrations compelling. I think the work is of broad interest, and suitable for publication in *eLife*, after addressing a few concerns and questions.

My comments below are mainly critical, but I note that I think the paper overall is very good, and thorough, and I have high enthusiasm.

Given that the surface profiles begin their optimization from an analytical expression, It would be nice if the authors could comment on the likely maximal possible correction possible – that is on a GRIN lens with a GRIN relay of significant pitch, do they think microlenses would be similarly manufacturable or effective?

In the simulated imaging data, especially the figure, and SNR conclusions drawn from that, I think it is important to more clearly identify the results as "simulations". The authors do label and describe the data as such, however, because of the look of the figure, and in the Discussion, it is very easy to misinterpret as measured experimental data, rather than an in silico sim – perhaps panel shading on a-h would help?

Regarding the simulations, it was not clear what the "imaging" rate was set to be. I am curious why processes where not included in the in silico simulation. I would expect the corrected FOVs to have even better performance than uncorrected, if processes were included, and it would have been nice to explore. Further, I am not quite sure why the authors did not use a standard method to extract the putative ROIs, and compare that with their ground truth – it seems they considered the flaws of standard practice, and biased their sampling of ROIs directly – but I would think using a standard pipeline instead would have been more informative.

Along with this, in the real experiments, different methods were used to pick the ROIs – I am wondering if the authors could comment on the rational for the choices? I say this also, because while one would prefer to record more faithful signals directly, it isn't clear to me that some of the additional "mixing" that would occur between the ROIs would not be removed through source factorization.

The slow imaging rate also leads to higher correlations. Did the authors also collect signals at high frame rates? The basic optical properties would not change at higher rates, but 2-3 Hz is far from typical functional imaging rates. The gains in SNR may be more material at these higher frame rates too. Was it simply a microscope limitation that prohibited faster imaging?

For Figure 7, It is not clear to me why there should be a bias toward information away from the center assuming (semi) random placement of the GRIN- do the authors think that this is real, or simply a measurement artifact – the center of the FOV is nominally furthest from completely healthy tissue, so the slightly lower information reflects network damage? That would also affect the magnitude (but likely not the fundamental interpretation) of the spatial distribution of cells in the modules.

Reviewer #3:

In this paper, the authors describe a novel approach for improving two-photon imaging through thin GRIN-lens endoscopes by using a nanofabricated corrective lens. They validate the approach with in vitro measurements and simulations, and apply the technique to imaging a variety of samples, most notably somatosensory thalamus in awake, behaving mice.

The technique seems easy to apply and could advance the state of the art in several labs. Although I haven't done such experiments myself, I know from other groups that experiments are often sorely limited by their field of view. This paper presents an exciting solution to this problem.

I am enthusiastic about the work, but point out the following issues:

1) With uncorrected GRIN endoscopes, 3D imaging is possible. Do the corrective lenses affect the axial range that can be scanned this way, and if they do, how does this impact the total of neurons that can be recorded with a corrected vs uncorrected endoscope? Zemax simulations would be sufficient to address this.

2) I think the paper could do a better job of motivating the need for thinner endoscopes, particularly in the Introduction. Figure 5 leaves me wondering whether connections would really be disrupted by a larger endoscope. Figure 3—figure supplement 7 makes it clear that larger endoscopes and cannulae require removing a lot of structures, but the paper leaves it to the reader to imagine the consequences. Have there been published accounts of failure rates for different endoscope sizes? Perhaps your own experiences, or studies of e.g. amygdala or hypothalamus?

3) The methods used to segment/analyze calcium imaging data (subsection “Segmentation of simulated time series”) come off as somewhat straw-man. There are a variety of activity-based tools that can separate mixed signals from overlapping cells (PCA/ICA, NMF, CNMF, etc.), and these have been popular for (1P) endoscopic imaging. I wonder if such methods narrow the gap between uncorrected and corrected SNRs and correlations. The statements at the end of paragraph five of the Discussion seem a bit strong if this possibility has not been explored.

4) How chromatic are these corrections? The authors propose simultaneous functional imaging and optogenetic perturbations with these corrective lenses, but this would require the correction be useful across a large wavelength range (perhaps ~900nm and ~1040nm), which needs more evidence. Chromatic issues can easily affect 2P imaging even over the bandwidth of a femtosecond laser.

---

## [Author Response]

Reviewer #1:In this manuscript, the authors demonstrated an extended field-of-view (eFOV), thin (<500μm) microendoscopes by adding aberration correcting microlenses fabricated bt TPL. Performance of the microlens corrected eFOV endoscopes was demonstrated with simulations on synthetic calcium data and in vivo activity imaging. In addition, the eFOV microendoscopes were used to investigate VPM activity correlated with locomotion, whisker movement, and pupil diameter. Cell-specific encoding in VPM during these behaviors are further scrutinized with statistical and machine learning methods. The presentation of the methods and the results are clear. The methods are useful and reasonably practical for increasing the FOV of ultrathin microendoscopes, which is a major limitation for small GRIN lenses. Overall, I recommend publication of the paper in eLife. Listed below are a number of concerns that we hope the authors can address in their revision to further improve the manuscript.1) For the design and fabrication of the eFOV-microendoscope:a) The authors presented four types of GRIN lenses (Type I-IV). Four different models of the GRIN lens are shown in subsection “Corrective lens manufacturing and microendoscope assembly”. It will probably help if the authors can explicitly indicate the Types with the model number.

The correspondence between the type and the model number of the GRIN lenses used in the study is now indicated in paragraph two of subsection “Corrective lens manufacturing and microendoscope assembly”.

b) It is claimed that the experimental result is similar to ray-trace simulation as shown in Figure 2. However, there is no comparison to support the claim. The high-order coefficient in the gradient profile in GRIN lens is usually not included in the optical model used for ray-trace. High-order aberration will thus not be reflected in the simulation. It would be important to know the difference between the simulation result and the experimental result. For example, the authors can add experimental data in Figure 2 or add simulation data in Figure 3.

Following the reviewer’s comment, we ran simulations (now shown in new Figure 2—figure supplements 1-2) to quantify the excitation PSF for the four types of endoscopes and to compare these simulation results with the experimental findings displayed in Figure 3. Similarly to what observed in real experiments, in optical simulations we found that the axial dimension of the PSF in lateral portions of the FOV remained smaller and more similar to the axial dimension of the PSF in the center of the FOV in corrected compared to uncorrected endoscopes. Thus, these new optical simulations predicted enlarged FOV in corrected microendoscopes, a result that we experimentally validated (Figure 3). We edited the text in paragraph two of the Results and figures (new Figure 2—figure supplements 1-2) to include these new findings.

It must be noted that the absolute values of the axial PSF were generally smaller in the optical simulations compared to real measurements. This may be due to multiple reasons. First, as suggested by the reviewer, high-order aberrations were not included in the simulations. Second, in simulations, although the intensity of the excitation PSF was small in lateral portion of the FOV (Figure 2), a Gaussian function could still reliably fit the dim intensity distribution and provide a clear quantification of the PSF dimension. In experimental measurements of fluorescence emitted by subresolved beads, the more degraded PSF in the lateral portions of the FOV would result in low efficacy of the excitation beam in stimulating fluorescence, which would result in low SNR fluorescence signals. This would introduce large variability in the fit. Third, small variability in some of the experimental parameters (e.g., the distance between the GRIN back end and the focusing objective) were not reflected in the simulations.

c) For all the experiments, is it performed with the 2-photon polymerized lens with resin or with lens replica using a molded UV-curable adhesive? Is the aberration correction performance different from these two fabrication methods? Which material is the ray-tracing simulation based on?

Experiments and optical characterization were performed using lens replica only. Ray-trace simulation were performed considering the material used in lens replica (i.e., NOA63). We edited the text to make this clearer.

d) What is the yield of the lens fabricated? How is the cured UV-curable adhesive detached from the PDMS mold? During imaging sessions, was there any damage on the lens?

We thank the reviewer for their comments and we inserted the requested information in the text. Specifically:

“The yield for 3D printed lenses and lens replica was ~ 100 %.”

During molding, one drop of UV-curable adhesive was first put on a coverslip, which was then pressed against the mold. One side of the UV-curable adhesive was in contact with the mold, the other side was instead attached to the coverslip. After UV curing, by gently pulling the glass coverslip away the lens made of UV curable adhesive detached easily from the PDMS mold, while remaining firmly attached to the coverslip. This is now more clearly described in subsection “Corrective lens manufacturing and microendoscope assembly”.

We observed no appreciable damage on the lens over imaging sessions (now reported in subsection “Optical characterization of eFOV-microendoscopes”). In considering this, please note that laser beam is not directly focused on the polymer lens, but rather about 100 µm above it (Figure 1).

e) How does the correction microlens affect the focal length of the GRIN lens. In other words, what is the allowable change in working distance before and after the correction?

The simulated focal length in the absence and presence of the corrective microlens for the four different types of endoscopes is now reported in a novel table (new Supplementary file 1-table 2). The difference in focal length between uncorrected and corrected endoscopes is in the range 2-23 µm.

f) There appears to be a much larger field curvature with the corrected GRIN lens. It would be helpful to discuss the impact of this on functional imaging.

A short paragraph describing the impact of the larger field curvature of corrected endoscopes on functional imaging has been added in the Discussion. It reads: “Corrected endoscopes are characterized by a curved FOV. In the case of type II corrected endoscopes, the change in the z coordinate in the focal plane can be up to 75 µm (Figure 3). This z value is smaller for all other corrected endoscope types (Figure 3). The observed field curvature of corrected endoscopes may impact imaging in brain regions characterized by strong axially organized anatomy (e.g., the pyramidal layer of the hippocampus), but would not significantly affect imaging in regions with homogeneous cell density within the z range described above (< 75 µm for type II corrected microendoscopes).”

g) Instead of AO with an active element (e.g., SLM), one could also put a fixed lens with the desired curvature in the place of the SLM. Compared to the microlens approach, one needs to modify the excitation path somewhat, but the lens can be fabricated using standard techniques and may be more accessible to most research labs. It might be informative to the readers to comment on this alternative approach.

Following the reviewer’s comment, we modified the text to discuss this possibility. The new text reads: “A potential alternative to the approach describe in this study would be to place a macroscopic optical element of the desired profile in a plane optically conjugated to the objective back aperture along the optical path. This solution could have the advantage of being manufactured using more standard techniques. However, it would require significant change in the optical set-up, in contrast to the built-in correction method that we describe in the present study. Moreover, this macroscopic optical element would have to be changed according to the type of microendoscope used”.

2) For the data taken with the eFOV-microendoscope, there is ~2x improvement in FOV for type III lens (Figure 3g). However, in Figure 3—figure supplement 5C, the images before and after correction for type III lens is almost the same.

We substituted the images in Figure 3—figure supplement 5C with new ones that are more representative of the results shown in Figure 3G.

3) For the data simulation and analysis:a) Analysis was shown to correlate whisking, locomotion and cell-specific encoding. Figure 6C and 6D show that locomotion is usually correlated with a higher ΔF/F. Could this be a result of the motion artifact during locomotion from the head-constraint mouse?

We think it is unlikely that the observed increase in ΔF/F is due to motion artifacts during locomotion because: *i)* we implemented a motion correction procedure in our data analysis; *ii)* the effect of potential motion artefacts on ΔF/F values would likely be that of increasing dispersion rather than biasing the mean towards higher values, as observed in Figure 6F. This is because a change in focus or a x,y translation due to a motion artefact would likely have similar probability of increasing or decreasing ΔF/F values.

b) To simulate synthetic calcium data, different models (Gaussian Mixture, lognormal, etc.) are used to obtain the pixel intensity value as described in subsection “Generation of fluorescence time series”. It is mentioned that the model is obtained by fitting the experimental data. What criteria (e.g. comparing likelihood) is used to choose those specific models? What other distribution is tested to choose the best-fit model?

We fitted the experimental data using four alternative models: a normal, a γ, a log-normal distribution, and a Gaussian mixture model. We selected the best-fit model by maximizing the log-likelihood.

In the following tables, the log-likelihood values of the used distributions pixels in the center and lateral portion of the FOV are reported for the reviewer’s inspection (highlighted in green the one we used under each condition):

Central pixels (without ROIs)Central pixels (with ROIs)

“*Edges” pixels (“dark noise”):*

c) As mentioned in subsection “Analysis of field distortion and calibration of the pixel size”, the author used a gaussian profile to define axial resolution. It is probably better to fit with a Lorentzian function I(z)=〖(1/(1+〖(z/zR)〗^2 ))〗^2 for 2-photon excitation, where zR is the Rayleigh range of the gaussian focus. Not sure how large the difference will be between these two fits.

Following the reviewer’s comment, we used the Lorentzian function to fit the fluorescence intensity distribution for type II microendoscopes. Axial resolution values computed using the fit with the Lorentzian function were not significantly different compared to those obtained fitting with a Gaussian function (Wilcoxon matched-pairs signed rank test, p = 0,93, n = 132 PSF measurements). We therefore decided to leave the results obtained with the Gaussian function in the manuscript.

d) Figure 4D. It is not clear to me why the ROI number is smaller for the corrected lens than the uncorrected when the peak SNR value is small (e.g., less than 10). Can the authors add some explanations?

We thank the reviewer for raising this point. The smaller number of ROIs when the SNR is low in corrected endoscopes is due to:

1) In the segmentation method we implemented, which is based on the ground truth distribution of the neurons in the simulated sample, at least two pixels belonging to a ground truth neuron are defined as a ROI.

2) In uncorrected endoscopes, the axial PSF largely increases as a function of the radial distance. The enlarged axial PSF in the lateral portions of the FOV augments the probability of sampling voxels belonging to multiple neurons located at different z positions. Once projected in the 2D plane, the contribution of multiple neurons located at different z positions increases the probability of having pixels belonging to ROIs. An increased axial PSF thus leads to an increased number of detected ROIs.

3) Corrected endoscopes have smaller axial PSF compared to uncorrected ones and thus smaller number of detected ROIs.

We added a short text to better explain this result in subsection “Higher SNR and more precise evaluation of pairwise correlation in eFOV-microendoscopes”.

e) The authors claim that "pairwise correlation increased as a function of the radial distance of the pair in uncorrected probes compared to corrected ones." Figure 4M does not seem to show noticeable increase as a function of the radial distance for the uncorrected lens (left panel). Rather, it seems to show a noticeable decrease as a function of the radial distance for the corrected lens (right panel). Therefore, the statement does not seem to be supported by Figure 4M. This is confusing, and some explanation is needed.

We thank the reviewer for raising this point and we apologize with them if the indicated sentence was imprecise and generated confusion. What we meant was that the linear fit of pairwise correlations as a function of radial position for uncorrected endoscopes had a significantly positive slope (Figure 4M, left panel, slope = 0.0002, permutation test p = 0.006), indicating higher pairwise correlations in lateral compared to more central portions of the FOV. For corrected endoscopes, the slope of the linear fit was not significantly different from zero (Figure 4M, right panel, slope = -0.0005, permutation test p = 0.05 for dataset 1; slope = -0.0007, permutation test p = 0.05 for dataset 2). We modified the text to correct the previous imprecision and clarify this point.

4) There is a typo in the first sentence of subsection “Fabrication of eFOV-microendoscopes”. I believe it is (Figure 3—figure supplement 1C) instead of (Figure 1C).

Thank you for spotting this error. It has been fixed.

Reviewer #2:In this paper, the authors describe the creation and use of microfabricated corrector lenses for improving GRIN lens imaging. GRIN lenses are one of a few technologies (and the only one that is "mature") that allow for high resolution optical imaging in brains at depths beyond ~1.5 mm. These correction lenses a) increase the usable field-of-view (FOV) of a given GRIN lens, and b) improve the fidelity of signals extracted from regions-of-interest (ROIs), by decreasing the point spread function (PSF) volume. Both are important for high fidelity recording of neuronal populations. The paper is well written, and I find their demonstrations compelling. I think the work is of broad interest, and suitable for publication in eLife, after addressing a few concerns and questions.My comments below are mainly critical, but I note that I think the paper overall is very good, and thorough, and I have high enthusiasm.Given that the surface profiles begin their optimization from an analytical expression, It would be nice if the authors could comment on the likely maximal possible correction possible – that is on a GRIN lens with a GRIN relay of significant pitch, do they think microlenses would be similarly manufacturable or effective?

We thank the reviewer for this interesting question. It is expected that optical aberrations of GRIN-rod lenses become more pronounced as the GRIN-rod length increases. This raises the possibility that the needed correcting lens profile could significantly differ from our current initial guess (the Schmidt corrector plate profile). To properly answer the reviewer’s question, there are thus a number of issues that need to be evaluated altogether, namely: a) up to what length of the GRIN-rod, for a given NA, the Schmidt corrector plate profile is an effective starting guess for the lens optimization process; b) once such an optimized correcting profile is found, to what extent the introduced correction is able to increase the effective FOV; c) how accurately the complex correcting lens profile could be reproduced by 3D micro-printing. Because our present approach satisfactory works for the GRIN-rod lengths considered in our study (1.1 mm – 4.07 mm), and because of the amount of work needed to address the manifold and interrelated aspects of the asked question, we deemed it to be the subject of a whole new study. As consequence, we currently cannot provide a final answer to this question, but we aim at addressing the matter in the continuation of our work. Please note that in the paper we do not make any claim that the method is applicable to GRIN lenses of any length, and we made sure in revision that this limitation in scope is acknowledged.

In the simulated imaging data, especially the figure, and SNR conclusions drawn from that, I think it is important to more clearly identify the results as "simulations". The authors do label and describe the data as such, however, because of the look of the figure, and in the Discussion, it is very easy to misinterpret as measured experimental data, rather than an in silico sim – perhaps panel shading on a-h would help?

We thank the reviewer for their suggestion. We followed the indication of the reviewer and modified Figure 4 to make clear, using boxes, which data come from simulations and which from real experiments.

Regarding the simulations, it was not clear what the "imaging" rate was set to be.

In the simulations the “imaging” rate was set to 5 Hz. This is indicated in subsection “Generation of fluorescence time series”.

I am curious why processes where not included in the in silico simulation. I would expect the corrected FOVs to have even better performance than uncorrected, if processes were included, and it would have been nice to explore.

We did not address the impact of including processes in the simulations, because the axial PSF of the GRIN microendoscopes was in the range 8-10 µm and the labeling that we achieved in our preparation was rather dense. Under these experimental conditions, resolving individual neuronal processes was complicated and we limited our goal to image fluorescent signals from neuronal cell bodies. We agree with the reviewer this is an interesting point, which can be addressed in future investigations.

Further, I am not quite sure why the authors did not use a standard method to extract the putative ROIs, and compare that with their ground truth – it seems they considered the flaws of standard practice, and biased their sampling of ROIs directly – but I would think using a standard pipeline instead would have been more informative.

Following the reviewer’s comment, we first compared the quality of the manual segmentation described in this study with that of a standard automated algorithm (e.g., CaImAn, (Giovannucci et al., 2019) by computing precision, recall, and F1 score in simulated data (new Figure 4—figure supplement 1). We found that the automated method was characterized by recall values which were, on average across SNR values, < 0.4 (new Figure 4—figure supplement 1A), leading to the detection of only a minority of the total number of neurons. In contrast, the manual method led to higher recall across SNR threshold values. Notably, both manual and automated methods yielded larger recall in corrected endoscopes compared to uncorrected ones. Moreover, for low SNR threshold values the automated segmentation had precision values < 0.8 in both uncorrected and corrected endoscopes (new Figure 4—figure supplement 1B), leading to identification of ROIs which did not correspond to cells in the ground truth. In contrast, the manual segmentation method had much larger values of precision across SNR threshold levels. Overall, F1 scores were higher for the manual segmentation method compared to the automated one for both uncorrected and corrected endoscopes (new Figure 4—figure supplement 1C). Because of these results showing better performance of the manual segmentations in simulated data (an observation now included in the revised manuscript), we took the decision to present in the main figures of the paper results based on manual segmentation.

In the revised manuscript, we extended the comparison between the manual and automated segmentation methods to real data. We observed that in uncorrected endoscopes CaImAn identified smaller number of ROIs compared to manual segmentation (new Figure 4—figure supplement 1D). In contrast, the number of ROIs identified with CaImAn and the manual method in t-series acquired with the corrected endoscope were not significantly different (new Figure 4—figure supplement 1E). One potential explanation of this finding is that the automated segmentation method more efficiently segments ROIs with high SNR compared to the manual one. Since aberration correction significantly increases SNR of fluorescent signals, the automated segmentation performed as the manual segmentation method in corrected endoscopes.

Overall, we believe that, in the context of our study, the use of the manual segmentation method rather than an automated segmentation algorithm is still preferable, because it allowed a more precise evaluation of ROIs in simulated data and fairer comparison of corrected and uncorrected endoscopes in real data. Most importantly, improvements introduced by corrected endoscopes could be observed with both the manual and the automated segmentation methods (see response to the next point). We believe that the comparisons suggested by reviewer are of interest and we feel that the new results in Figure 4—figure supplement 1 improve our manuscript. We are grateful to the reviewer for this suggestion.

Along with this, in the real experiments, different methods were used to pick the ROIs – I am wondering if the authors could comment on the rational for the choices? I say this also, because while one would prefer to record more faithful signals directly, it isn't clear to me that some of the additional "mixing" that would occur between the ROIs would not be removed through source factorization.

In all experiments presented in the main paper, we used our customized manual pipeline for the analysis of fluorescence t-series. In Figure 3—figure supplement 6 we used a different method, the PCA/ICA algorithm described in Mukamel et al., Neuron 2009, to segment neurons and extract calcium signals. This was originally done to show that data recorded with corrected endoscopes could be analyzed with standard automated methods. Following the comments of this reviewer and that of reviewer #3 asking for a systematic evaluation of how much the improvements introduced by aberration correction depend on the analysis pipeline, we removed the PCA/ICA analysis in Figure 3—figure supplement 6 and we systematically evaluated the effect of aberration correction on the output of the analysis in the simulated and experimental data shown in Figure 4 using both the manual segmentation method and an automated segmentation method (e.g., CaImAn).

In simulated data, we found that with CaImAn the number of ROIs segmented in corrected endoscopes was consistently higher than in uncorrected endoscopes across SNR thresholds (new Figure 4—figure supplement 2A), similarly to what observed with manual segmentation (Figure 4D). Using CaImAn, SNR values of fluorescence events were significantly higher in corrected compared to uncorrected endoscopes (new Figure 4—figure supplement 2B), similarly to what observed with manual segmentation (Figure 4E). Moreover, the linear fit of pairwise correlations as a function of radial position for uncorrected endoscopes had a significantly positive slope (new Figure 4—figure supplement 2C left panel), indicating higher pairwise correlations in lateral compared to more central portions of the FOV. For corrected endoscopes, the slope of the linear fit was not significantly different from zero (new Figure 4—figure supplement 2C right panel). This result is also in line with what we previously observed with manual segmentation (Figure 4G). The description of these novel results is added in the text.

In experimental data, we found that SNR values of fluorescence events tended to be higher in corrected compared to uncorrected endoscopes (new Figure 4—figure supplement 2D), a trend that was in line with what observed with manual segmentation (Figure 4L). The slope of the linear fit of pairwise correlations as a function of radial position for uncorrected endoscopes was significantly positive (new Figure 4—figure supplement 2E), indicating higher pairwise correlations in lateral compared to more central portions of the FOV. For corrected endoscopes, the slope of the linear fit was not significantly different from zero (new Figure 4—figure supplement 2E). Both results are in line with what previously observed with manual segmentation (Figure 4M). The description of these novel results is added in the text.

Overall, the results of the systematic comparison between the manual and automated segmentation methods show that improvements introduced by aberration correction in endoscopes could be observed with both the manual and the automated segmentation methods. We thank the reviewer for raising this point.

The slow imaging rate also leads to higher correlations. Did the authors also collect signals at high frame rates? The basic optical properties would not change at higher rates, but 2-3 Hz is far from typical functional imaging rates. The gains in SNR may be more material at these higher frame rates too. Was it simply a microscope limitation that prohibited faster imaging?

We did not collect fluorescent signals at frame rates higher than 4 Hz. This was because we aimed at imaging the largest possible FOV and our experimental set up was equipped with regular galvanometric mirrors and not with resonant mirrors. We added a sentence in the text to highlight this limitation of our study.

For Figure 7, It is not clear to me why there should be a bias toward information away from the center assuming (semi) random placement of the GRIN- do the authors think that this is real, or simply a measurement artifact – the center of the FOV is nominally furthest from completely healthy tissue, so the slightly lower information reflects network damage? That would also affect the magnitude (but likely not the fundamental interpretation) of the spatial distribution of cells in the modules.

At the single cell level, information was evenly distributed across the FOV (Figure 7C). We believe the reviewer’s comment refers to data shown in Figure 7E, which contains data pooled from a population of neurons. In Figure 7E, we observed an increase in the information carried by the population of neurons when we incrementally considered larger populations by adding ROIs at larger radial distances. This does not imply that ROIs away from the center carry higher information, but only that their information sums to the information of more central ROIs in a way that is not purely redundant. We apologize if the description of Figure 7E was confusing, and we edited the text to make it clearer.

Reviewer #3:In this paper, the authors describe a novel approach for improving two-photon imaging through thin GRIN-lens endoscopes by using a nanofabricated corrective lens. They validate the approach with in vitro measurements and simulations, and apply the technique to imaging a variety of samples, most notably somatosensory thalamus in awake, behaving mice.The technique seems easy to apply and could advance the state of the art in several labs. Although I haven't done such experiments myself, I know from other groups that experiments are often sorely limited by their field of view. This paper presents an exciting solution to this problem.I am enthusiastic about the work, but point out the following issues:1) With uncorrected GRIN endoscopes, 3D imaging is possible. Do the corrective lenses affect the axial range that can be scanned this way, and if they do, how does this impact the total of neurons that can be recorded with a corrected vs uncorrected endoscope? Zemax simulations would be sufficient to address this.

The optical simulations we did to design the corrective lenses were performed maximizing aberration correction only in the focal plane of the endoscope. Following the reviewer’s comment, we explored the effect of aberration correction outside the focal plane using new Zemax simulations. In corrected endoscopes, we found that the Strehl ratio was > 0.8 (Maréchal criterion) in a larger volume compared to uncorrected endoscopes (new Figure 2—figure supplement 3). The range of volume with high Strehl ratio in corrected endoscopes was 1.7-3.7 larger than the volume with high Strehl ratio in uncorrected probes, leading to similarly larger number of neurons being imaged in the enlarged volume. This is now described on lines (paragraph three subsection “Optical simulation of eFOV-microendoscopes”).

2) I think the paper could do a better job of motivating the need for thinner endoscopes, particularly in the Introduction. Figure 5 leaves me wondering whether connections would really be disrupted by a larger endoscope. Figure 3—figure supplement 7 makes it clear that larger endoscopes and cannulae require removing a lot of structures, but the paper leaves it to the reader to imagine the consequences. Have there been published accounts of failure rates for different endoscope sizes? Perhaps your own experiences, or studies of e.g. amygdala or hypothalamus?

Following the reviewer’s comment, we modified the Introduction to more clearly motivate the need of thin endoscopic probes for deep brain functional imaging in paragraph two of the Introduction. We are not aware of a systematic description on the impact of different endoscope sizes on the functionality of the targeted brain region, but we have cited relevant literature showing that surgical lesion of reciprocal connectivity alters thalamocortical dynamics.

3) The methods used to segment/analyze calcium imaging data (subsection “Segmentation of simulated time series”) come off as somewhat straw-man. There are a variety of activity-based tools that can separate mixed signals from overlapping cells (PCA/ICA, NMF, CNMF, etc.), and these have been popular for (1P) endoscopic imaging. I wonder if such methods narrow the gap between uncorrected and corrected SNRs and correlations. The statements at the end of paragraph five of the Discussion seem a bit strong if this possibility has not been explored.

Following the reviewer’s comment we evaluated the effect of aberration correction on the output of the analysis in the simulated and experimental data shown in Figure 4 using both the manual segmentation method and an automated segmentation method (e.g., CaImAn).

In simulated data, we found that with CaImAn the number of ROIs segmented in corrected endoscopes was consistently higher than in uncorrected endoscopes across SNR thresholds (new Figure 4—figure supplement 2A), similarly to what observed with manual segmentation (Figure 4D). Using CaImAn, SNR values of fluorescence events were significantly higher in corrected compared to uncorrected endoscopes (new Figure 4—figure supplement 2B), similarly to what observed with manual segmentation (Figure 4E). Moreover, the linear fit of pairwise correlations as a function of radial position for uncorrected endoscopes had a significantly positive slope (new Figure 4—figure supplement 2C left panel), indicating higher pairwise correlations in lateral compared to more central portions of the FOV. For corrected endoscopes, the slope of the linear fit was not significantly different from zero (new Figure 4—figure supplement 2C right panel). This result is also in line with what previously observed with manual segmentation (Figure 4G). The description of these novel results is added in subsection “Higher SNR and more precise evaluation of pairwise correlation in eFOV-microendoscopes”.

In experimental data, we found that SNR values of fluorescence events tended to be higher in corrected compared to uncorrected endoscopes (new Figure 4—figure supplement 2D), a trend that was in line with what observed with manual segmentation (Figure 4L). The slope of the linear fit of pairwise correlations as a function of radial position for uncorrected endoscopes was significantly positive (new Figure 4—figure supplement 2E), indicating higher pairwise correlations in lateral compared to more central portions of the FOV. For corrected endoscopes, the slope of the linear fit was not significantly different from zero (new Figure 4—figure supplement 2E). Both results are in line with what previously observed with manual segmentation (Figure 4M).

The description of these novel results is added in subsection “Higher SNR and more precise evaluation of pairwise correlation in eFOV-microendoscopes”.

Overall, the results of the systematic comparison between the manual and automated segmentation methods show that improvement introduced by aberration correction in endoscopes could be observed with both the manual and automated segmentation methods. We thank the reviewer for raising this point.

4) How chromatic are these corrections? The authors propose simultaneous functional imaging and optogenetic perturbations with these corrective lenses (Discussion final paragraph), but this would require the correction be useful across a large wavelength range (perhaps ~900nm and ~1040nm), which needs more evidence. Chromatic issues can easily affect 2P imaging even over the bandwidth of a femtosecond laser.

Thanks for this comment. All optical simulations described in the first submission were performed at a fixed wavelength (λ = 920 nm). Following the reviewer’s request, we explored the effect of changing wavelength on the Strehl ratio using new Zemax simulations. We found that the Strehl ratio remains > 0.8 within ± 15 nm from λ = 920 nm (new Figure 2—figure supplement 4), which covers the limited bandwidth of our femtosecond laser but not that needed for all-optical imaging and manipulation experiments. To perform this latter type of experiments, the SLM that is used in the stimulation pathway should also be used to correct for the z-defocus with an appropriate lens function. Alternatively, future microfabrication work will be needed to design more complex corrective lens design, which efficiently compensates chromatic aberrations in the 900-1100 nm wavelength range. These possibilities are now more clearly discussed (Discussion paragraphs eight and nine).